# Scalarization for Multi-Task and Multi-Domain Learning at Scale

**Amélie Royer, Tijmen Blankevoort, Babak Ehteshami Bejnordi**
Qualcomm AI Research[*]
Amsterdam, The Netherlands
`{aroyer, tijmen, behtesha}@qti.qualcomm.com`

## Abstract

Training a single model on multiple input domains and/or output tasks allows for compressing information from multiple sources into a unified backbone hence improves model efficiency. It also enables potential positive knowledge transfer across tasks/domains, leading to improved accuracy and data-efficient training. However, optimizing such networks is a challenge, in particular due to discrepancies between the different tasks or domains: Despite several hypotheses and solutions proposed over the years, recent work has shown that uniform scalarization training, i.e., simply minimizing the average of the task losses, yields on-par performance with more costly SotA optimization methods. This raises the issue of how well we understand the training dynamics of multi-task and multi-domain networks. In this work, we first devise a large-scale unified analysis of multi-domain and multi-task learning to better understand the dynamics of scalarization across varied task/domain combinations and model sizes. Following these insights, we then propose to leverage population-based training to efficiently search for the optimal scalarization weights when dealing with a large number of tasks or domains.

## 1 Introduction

Learning a unified architecture that can handle multiple domains and/or tasks offers the potential for improved computational efficiency, better generalization, and reduced training data requirements. However, training such models proves a challenge: In particular, when the tasks or domains are dissimilar, practitioners often observe that the learning of one task or domain interferes with the learning of others. This phenomenon is commonly known as interference or negative transfer. How to best resolve this interference is an open problem spanning numerous bodies of literature, such as multi-task learning, domain adaptation and generalization, or multi-modal learning.

In fact, there is currently no unanimous understanding of what causes, or how to predict, task interference in practice: For instance, classical generalization bounds on training from multiple data sources [3, 48, 56, 44] involve a measure of distance between the respective tasks/domains distributions, which can be intuitively thought of as a measure of interference. In more practical applications, task clustering methods [65, 58, 16] propose various metrics to measure task affinity, and train tasks with low affinity separately, with separate backbones, thereby circumventing the interference issue at the architecture level. Finally, in the multi-task optimization (MTO) literature, the prevailing view is that gradients from different tasks may point in conflicting directions and cause interference, which has led to a line of work on reducing gradient conflicts by normalizing gradients or losses statistics [37, 7, 25, 64]. While MTO have established a new state-of-the-art for training multi-task models in the past few years, recent work [63, 30] shows that, surprisingly, minimizing

---

[*] Qualcomm AI Research is an initiative of Qualcomm Technologies, Inc.

a simple average of the tasks/domains losses, also known as *scalarization*, can yield performance trade-off points on the same Pareto front as more costly MTO methods.

Nevertheless, there still remains some unexplored areas which we aim to shed lights on in this paper. First, most previous experiments have only been conducted on multi-task settings, on benchmarks with either few tasks or few training samples (e.g multi-MNIST, NYU-v2), and for a fixed architecture. In particular, the link between model capacity and the observed MTL performance is often overlooked. Secondly, it is not clear how the choice of scalarization weights links to and/or impacts the hypothesis of gradient conflicts as a key underlying signal of task interference. Finally, scalarization can become prohibitively expensive for a large number of tasks as the search space for the optimal loss weights grows exponentially; This raises the issue of how to efficiently browse this search space. In this work, we investigate these questions to further motivate scalarization as a simple and scalable training scheme for multi-task and multi-domain problems.

**Our key contributions.** We perform a large-scale analysis of scalarization for both multi-task (MTL) and multi-domain learning (MDL). We cover a wide range of model capacities, datasets with varying sizes, and different task/domain combinations. Our key conclusions are as follows:

- **(C1)** When compared to a model trained on each task/domain individually, MDL/MTL performance tends to improve for larger model sizes, showing that the benefits of MTL/MDL frameworks should be put into perspective with respect to the backbone architecture capacity.

- **(C2)** Tuning the scalarization weights for a specific tasks/domains combination is crucial to obtaining the optimal MTL/MDL performance in settings with high imbalance; Nevertheless, the relative performance of different scalarization weights is often consistent across model capacities inside a family architecture. This suggests that searching for optimal scalarization weights for a lower model depth/width is also relevant for the full model size, while using less compute.

- **(C3)** Gradients conflicts between tasks/domains naturally occur during MTL training: They behave differently across layers and learning rates, but are scarcely impacted by model capacity and scalarization weights choice. These observations give new insights into the practical implications of avoiding conflicting gradients throughout training.

- **(C4)** We leverage fast hyperparameter search methods such as population-based training [24] to efficiently browse the search space of scalarization weights as the number of tasks grows.

## 2 Related work

**Multi-Task Optimization (MTO) and scalarization.** MTO methods aim to improve MTL training by balancing the training dynamics of different tasks. Prior research can be split into two categories: On the one hand, loss-based methods propose to align the task losses magnitudes by rescaling them through various criteria, e.g., task uncertainty [26], task difficulty [21], random loss weighting [34], or gradients statistics [7, 14, 60]. On the other hand, gradient-based methods directly act on the per-task gradients rather than the losses. For instance, [55, 13] tackle MTL as a multi-objective optimization problem using the Multiple Gradient Descent Algorithm to directly locate Pareto-optimal solutions. Another major line of work considers conflicting gradient direction to be the main cause of task interference. Consequently, these works [64, 37, 38, 62, 8, 25] aim to mitigate conflicts across per-task gradients. For instance, PCGrad [64] suggests projecting each task's gradient onto the normal plane of other gradients to suppress conflicting directions, while GradDrop [8] ensures that all gradient updates are pure in sign by randomly masking all positive or negative gradient values during training. Unfortunately, gradient-based techniques require per-task gradients, leading to substantial memory usage and increased runtime. Furthermore, recent work [63, 30] has shown that these methods often perform on-par with the less costly approach of directly optimizing the average of the task losses, also known as uniform scalarization. Building off these insights, we further investigate the benefits of scalarization and the practical dynamics of gradient conflicts in this work.

**Architectures for MTL.** Orthogonal to our work, another line of research focuses on designing multi-task architectures that mitigate task interference by optimizing the allocation of shared versus task-specific parameters. Hard parameter sharing methods [27, 26, 6, 59, 61] build off a shared backbone and carefully design task-specific decoder heads optimized for the tasks of interest. In contrast, soft parameter sharing methods [45, 53, 20, 39] jointly train a shared backbone while learning sparse binary masks specific to each task/domain which capture the parameter sharing pattern across tasks. To select which tasks should share parameters or not, a prominent line of work

focuses on defining measures of task affinity [2, 58, 22, 16]: Tasks with high affinity are jointly trained, while low affinity ones use different backbones, in the hope of reducing potential interference by fully separating conflicting tasks. This results in multi-branch architectures where each branch handles a specific subset of tasks with high affinities. The scope of our study is complementary to this line of work, as we focus on how to best train a given set of tasks, without changing the architecture.

**Multi-Domain Learning (MDL).** MDL refers to the design and training of a single unified architecture that can operate on data sources from different domains. Domain adaptation and generalization methods often aim to align the learned representations of the different domains [19, 4, 66]; These are in particular targeting the setting where one of the domain lacks supervisory signals. In the fully-supervised scenario, several methods have been proposed to use a unified backbone, which captures common features across all tasks, while allowing the model to learn domain-specific information via lightweight adapter modules [49, 51, 50]. Similar to soft parameter sharing method, these methods bypass task interference by keeping the shared backbone frozen and only training domain-specific modules. Alternatively, a common practice when training with multiple input datasets is to use over- or undersampling techniques [1, 28, 43], in particular to handle class imbalance. As we describe later, resampling can be seen as the MDL counterpart of scalarization in MTL. Both enable a simple and lightweight training scheme, yet finding the optimal resampling/scalarizatoin weights is nontrivial. This motivates us to empirically verify our findings in the MDL as well as MTL setting.

## 3 Motivation and experimental setting for MTL/MDL

### 3.1 Notations

We start by describing the MTL/MDL setup: Given $T$ supervised datasets with respective inputs $X_t$ and outputs $Y_t$, our goal is to find model parameters $\theta^*$ that minimize the datasets' respective losses: $(\mathcal{L}_1(X_1, Y_1), \ldots, \mathcal{L}_T(X_T, Y_T))$. In practice, this can be achieved either by solving a multi-objective problem on the task losses [13, 42] or, more commonly, by minimizing a (possibly weighted) average of the losses [27]. In both cases, most training methods for MTL/MDL can be phrased as updating the model parameters $\theta$ using a weighted average of the individual tasks' gradients:

$$\theta_{i+1} = \theta_i + \eta \sum_{t=1}^{T} p_t^i \, \mathbb{E}_{(x_t, y_t) \sim q(X_t, Y_t)} \nabla_{\theta_i} \mathcal{L}_t(x_t, y_t) \, , \text{ where } \eta \text{ is the learning rate} \qquad (1)$$

where $\mathbf{p}(\mathbf{i}) = (p_1^i, \ldots, p_T^i)$ captures the dynamic importance scaling weights for each dataset at timestep $i$, and $q(X_t, Y_t)$ is the underlying distribution the $t$-th dataset is drawn from. We also follow the common assumption that $\mathbf{p}$ is a probability distribution, *i.e.*, $\sum_t p_t = 1$ and $\forall t, \ p_t \in \{0, 1\}$.

We further distinguish between **(i)** multi-task learning (MTL) where every input is annotated for every task (i.e., $\forall i, j, X_i = X_j$), and **(ii)** multi-domain learning (MDL) where the goal is to solve a common output task across multiple input domains or modalities ($\forall i, j, Y_i = Y_j$). In the MDL setting, we can also rephrase (1) as resampling the data distributions according to $\mathbf{p}$. Both formalisms are equivalent, but resampling often performs better in practice when using stochastic gradient methods [1]:

$$\underbrace{\sum_{t=1}^{T} p_t \mathbb{E}_{q(X_t, Y_t)} f(x_t, y_t)}_{\text{Reweighing formalism of (1)}} = \underbrace{\mathbb{E}_{q'(X,Y)} f(x, y)}_{\text{Resampling formalism}} \text{ where } q'(x, y) = \underbrace{\sum_{t=1}^{T} \mathbb{1}_{x \in X_t} q(x, y) p_t}_{\text{Resampling distribution}} \qquad (2)$$

### 3.2 Motivation

Current state-of-the-art methods for training objectives such as (1) can be roughly organized into three categories, based on how the datasets' importance weights $\mathbf{p}$ are defined:

- **Scalarization** defines the weights $p_t$ as constant hyperparameters: The update rule of (1) reduces to computing the gradient of the weighted average loss $\nabla_\theta \left( \sum_t p_t \mathcal{L}_t \right)$. In the MDL formalism of (2), scalarization is similar to classical oversampling/undersampling, where $p_t$ is defined either as a scalar (per-domain) or a vector (to handle both per-class and per-domain biases). In both settings, the key difficulty lies in tuning the weights $\mathbf{p}$, as the search space grows exponentially with $T$.

- **Loss-based adaptive methods** [38, 26] dynamically compute $p_t$ for every batch, aiming to uniformize training dynamics across tasks by rescaling the losses. For instance, IMTL-L [38] dynamically reweighs the losses such that they all have the same magnitude.

- **Gradient-based adaptive methods** [7, 64, 38, 25, 8, 37, 13, 55, 36, 42] also dynamically compute weights for every batch. While this line of work usually outperforms loss-based methods, they also incur a higher compute and memory training cost, as they require $T$ backward passes to obtain each individual dataset gradient, which in turn need to be stored in memory.

While gradient-based approaches are generally considered SotA across MTO methods, they present certain practical challenges: They come with a higher computational and memory cost, which increases with $T$, as illustrated in Figure 1(b). The implicit weights $\mathbf{p}_t$ computed by these methods are also hard to extract and transfer to other training runs since they are tied to other hyperparameters (training length, batch size, gradient clipping, etc.). In addition, recent work [63, 30, 52] has shown that simple scalarization with uniform weights actually often performs on-par with both loss- and gradient-based methods. Furthermore, scalarization is highly practical: It does not incur additional costs during training, and scalarization weights are easily interpretable as a measure of "task importance" in the optimization problem. However, some experiments in [58, 52] also suggest that the benefits of uniform scalarization MTL are impacted by model capacity in some settings.

Motivated by these insights, we aim to better harness scalarization for scalable and practical MTL/MDL training. In particular, we focus on the following two remaining gray areas: **First**, we perform a large-scale analysis investigating the underlying effects of model capacity on MTL and MDL *(C1)* in Section 4. We then discuss the impact of model capacity on the choice of optimal scalarization weights *(C2)* in Section 4.1 and on the presence of conflicting gradients *(C3)* in Section 4.2. **Secondly**, browsing the search space of $\mathbf{p}$ to tune the scalarization weights becomes increasingly costly as the number of tasks $T$ grows. As an alternative to the usual grid- or random-search approaches, we propose to leverage population-based training as a cost-efficient parameter search for scalarization, and report our conclusions *(C4)* in Section 5.

|  | MDL | | MTL | |
|---|---|---|---|---|
| Dataset | CIFAR+STL [29, 9] | DomainNet [47] | CelebA [40] | Taskonomy [65] (tiny split) |
| # tasks or domains, $T$ | 2 | 6 | 8 or 40 | 7 |
| Training size, $\|X\|$ | 55k | $\sim 410$k | $\sim 162$k | $\sim 275$k |
| Backbone | ViT-S/4 | ResNet-101 | ViT-S/4 | ResNet-26 encoder/decoder |
| Depth sweep | {3, 6, 9} | {r26, r50, r101} | {3, 6, 9} | {4, 2, 0} shared decoder layers |
| Width sweep | {0.25, 0.5, 0.75, 1.0} | {0.25, 0.5, 1.0} | {0.25, 0.5, 0.75, 1.0} | { 0.5, 1.0} |

**(a)** Summary of our experimental setting

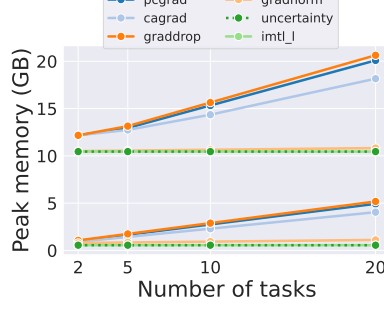

**(b)** Memory usage of various multi-task optimization (MTO) methods

Figure 1: Table **(a)** summarizes the experimental setups used throughout the paper. Figure **(b)** reports profiling results for popular multi-task optimization (MTO) methods in a small-scale (ResNet18, batch size 16) and large-scale (ResNet50, size 128) setting, for 224px inputs: In practice, we find that the main bottleneck for gradient-based methods is their high memory usage. Consequently, training with reasonable batch sizes requires either high parallelism (compute demand) or gradient accumulation (slower runs). In contrast, loss-based methods have much better-constrained costs. However, they usually underperform their gradient-based counterpart in the literature.

## 3.3 Experimental setting

Here we briefly describe the experimental setup that is used throughout the paper. We aim to cover a wide range of model sizes, datasets with varying sizes, and different task/domain combinations for MDL and MTL. A summary is given in Figure 1(a) and further details can be found in Appendix 1.

For MDL, we first consider a two-domain example composed of CIFAR10 [29] and STL10 [9] with a ViT-S backbone optimized for small datasets [18]. We then expand the results to the DomainNet

benchmark [47] containing 345 output classes and 6 input domains. We use ResNet as our main backbone, following previous works [47, 32]. In both cases, the backbone parameters are fully shared across all domains; We use cross-entropy as training loss and top-1 accuracy as our main metric.

For MTL, we use CelebA [40] with the same ViT-S backbone as for CIFAR/STL; We split the 40 attributes into semantically-coherent groups to form 8 output classification tasks, as described in the appendix. The transformer backbone is fully shared across tasks, while the last linear classification layer is task-specific. Then, we experiment on a larger MTL setting for dense prediction: Most traditional benchmarks, such as Cityscapes [10] and NYU-v2 [57], are rather small, as they only contain 2-3 dense tasks (segmentation, normals, depth) and $\sim$ 5k fully annotated images. Instead, we perform our analysis on the challenging Taskonomy dataset [65]. To keep experiments scalable, we filter the 26 tasks in Taskonomy down to 7 (e.g. by removing self-supervised tasks or clearly overlapping ones such as 2D edges and 3D edges). For training, we follow the setup described in [58]: We normalize every dense annotation to have zero mean and unit variance across the training set, and use the $L_1$ loss as our training loss and common metric for all tasks. The backbone is composed of a (shared) ResNet encoder, followed by task-specific decoders built with upsampling operations and 1x1 convolutions. To control model capacity, we vary *(i)* the width of the model, *(ii)* the depth of the encoder, as well as *(iii)* the number of layers in the decoder(s) which are shared across tasks.

Finally, in all settings, we sweep over the domains/tasks' weights under the common assumption that $p$ is a probability vector (i.e. $\forall t,\ p_t \in \{0, 1\}$ and $\sum_t p_t = 1$). All reported results are averaged across 2 random seeds for DomainNet and Taskonomy, and 3 for the smaller benchmarks. Unless stated, every experiment is conducted on a single NVIDIA V100 GPU.

# 4 Benefits of MDL/MTL under the lens of model capacity

We first investigate the behavior of scalarization for MDL/MTL training with two tasks, while varying model capacity and for different tasks/domains combinations. To quantify the benefit or harm of joint training with multiple datasets, we compare the model performance on each dataset with that of a same-sized model, trained on a single dataset at once. We refer to this baseline as "SD" (single dataset). Since we use SD as a reference point, we ensure that its training pipeline is well tuned: We sweep over the hyperparameters (learning rate, weight decay, number of training steps, etc) of the baseline SD and keep the same values for training the MDL/MTL models, while varying the task weights $(p_1, p_2 = 1 - p_1)$ from $p_1 = 0$ (SD baseline of the first dataset) to $p_1 = 1$ (SD baseline of the second dataset). Further details on the training hyperparameters can be found in Appendix 1. Finally, we summarize our analysis results in Figure 2: For each task/domain pair, we first plot the weight $p_1^*$ yielding the best accuracy averaged across both tasks/domains, then we report the accuracy difference between the corresponding MDL/MTL model and the associated SD baseline.

**Impact of model capacity.** We first observe that MDL/MTL performance greatly varies across model capacities, and tends to increase with model size *(C1)*: In some cases, the trend even inverts from MDL/MTL underperforming to outperforming SD (e.g. DomainNet's real + sketch pairing).

**Selecting optimal importance weights p.** Secondly, the best-performing **p** at inference are rarely the uniform $p_1 = p_2 = 0.5$, even in the Taskonomy scenario where the uniform training objective is identical to the average of task metrics which we aim to maximize at inference. We further discuss the effects of tuning scalarization weights for the accuracy/efficiency trade-off *(C2)* in Section 4.1.

**MTL/MDL primarily improves generalization.** As we show in Appendix 2), even when MTL/MDL outperforms the SD baseline at inference, it is generally not the case at training time. In other words, tuning the respective tasks/domains weights in MTL/MDL can be seen as a regularization technique that improves generalization over the SD baselines by balancing the two datasets' training objectives. In particular, in the MDL setting, this insight is also closely connected to data augmentations: Each new domain can be seen as a non-parametric and non-trivial augmentation function, while the associated resampling weight $p_t$ is the probability of applying the augmentation on each sample.

Interestingly, recent work [30] suggests that many MTO methods which aim to avoid gradient conflicts can also be interpreted as regularization techniques, and that simpler tricks such as early stopping are often competitive. To better understand the link between these two insights, we analyze the natural emergence of conflicting gradients during MDL/MTL training, and how they are impacted by model capacity *(C3)* in Section 4.2.

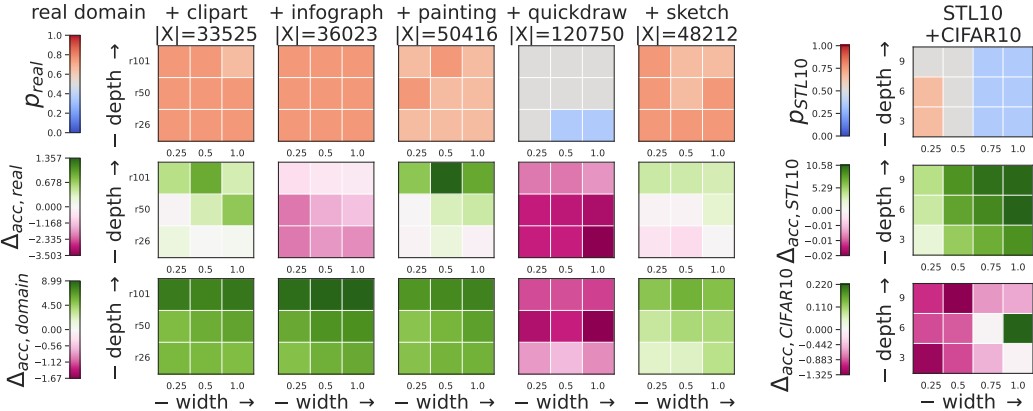

**(a)** Analysis of MDL when pairing natural images (`real` domain) with each of the five remaining DomainNet domains (left), and on the CIFAR-10 + STL-10 toy example (right)

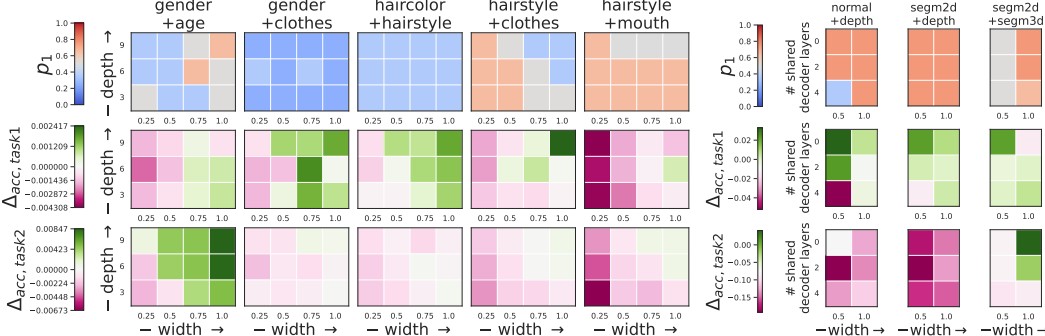

**(b)** Analysis of MTL on CelebA (left) and Taskonomy-tiny (right) for different task pairs and model sizes.

Figure 2: Performance of scalarization for MDL/MTL relative to SD under different model capacities; Each column corresponds to a different task/domain pair ($T = 2$). The first row of each plot contains a heatmap of the best performing scalarization weights $p^*$ wrt. to the average test accuracy on both tasks. Each of the following rows contains the difference in metrics between the MTL/MDL model and its counterpart SD baseline, where green indicates positive changes and purple, a negative one. Note that the colormaps' ranges are defined *per row*, and visualized as color bars at the beginning of each row. We observe the general trend that the performance improvement of MDL, relative to the corresponding SD baseline, tends to increase with model capacity.

## 4.1 Selecting optimal scalarization weights $\mathbf{p}^*$

So far, we have discussed MDL/MTL improvement over the SD baseline when selecting the weights $p^*$ that maximize average accuracy. However, in real-world settings, many criteria come into play when evaluating MTL/MDL models: For instance, one task may be more critical than others hence the model should be evaluated via weighted accuracy. Second, even if the MTL/MDL model underperforms the corresponding SD baseline at the same model capacity, it may still improve the efficiency-to-accuracy trade-off when both tasks need to be solved at once. To give a clearer overview of how these considerations affect the choice of weights **p**, we report results for all scalarization weights in Figure 3; We also highlight points where MTL/MDL outperforms SD on *both* domains/tasks, and represent each model's number of parameters as the marker size.

First, we observe a clear asymmetry in terms of performance across the $T = 2$ datasets, even when using uniform weighing $p_1 = 0.5$: For instance on Taskonomy (Figure 3), several MTL models outperform SD on the depth prediction task, but none on the semantic segmentation task. Nevertheless, MTL models are very appealing when taking model efficiency into account, as they contain roughly half as many parameters. Secondly, tuning the scalarization weights is crucial in some settings: For instance in DomainNet, training with $p_1 \in [0.65, 0.75]$ is more advantageous than uniform scalarization. Furthermore, the relative ranking of the weight $p_1$, with respect to model performance, does not change significantly across model capacities, for both the Taskonomy and

DomainNet examples. This suggests that optimal weights $\mathbf{p}^*$ for a given model may be a good search starting point for another architecture of the same family (see Appendix 4).

Nevertheless, the search space for $\mathbf{p}$ grows exponentially with $T$, making the search computationally prohibitive, even for one architecture. In Section 5, we propose a scalable approach to optimize scalarization weights and investigate its performance on DomainNet and CelebA.

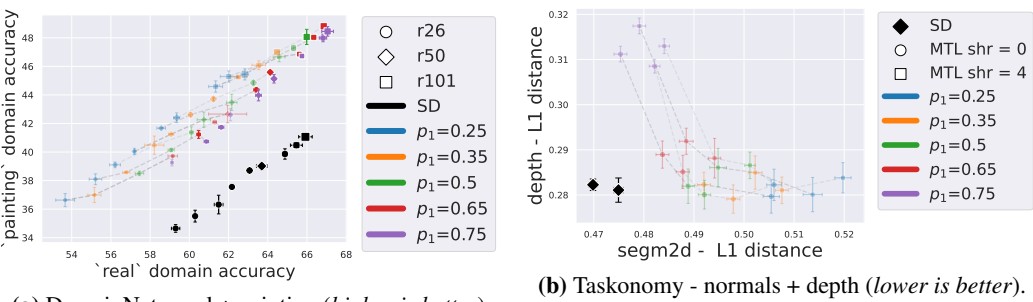

(a) DomainNet - real + painting (*higher is better*)

(b) Taskonomy - normals + depth (*lower is better*). "shr" refers to the number of shared decoder layers.

Figure 3: MDL/MTL performance when varying tasks weight $p_1$ for different model capacities. High opacity markers represent models that outperform their respective SDL baseline (black markers) on *both* tasks/domains. Dashed lines connect models with the same architecture. The marker size is proportional to the number of parameters in each model.

## 4.2 Conflicting gradients in practice

A widely spread explanation for task interference in the literature is that individual task gradients may point in conflicting directions, hampering training. To investigate this behavior, we measure the percentage of conflicting gradients pair encountered in each epoch, when training with uniform scalarization. Following [64], we define gradients as conflicting if and only if the cosine of their angle is negative. Finally, we provide further details and figures for this section in Appendix 3.

In Figure 4, we first illustrate an asymmetric characteristic of gradient conflicts: A high number of gradient conflicts typically translates to poorly performing models (e.g. early training), but the lower conflicts regime does not correlate well with MDL/MTL performance. In particular, it is common to encounter more gradient conflicts towards the end of training, while the loss steadily decreases. This suggests that, in practice, identifying and removing conflicts at every training iteration may be superfluous, with respect to the compute and memory cost occurred. We can also put this observation in perspective with Theorem 3 of [64], which states that in the case of two tasks, a parameter update of PCGrad leads to a lower loss than the uniform scalarization update if the tasks are conflicting enough; however, this assumption may not hold true across every training iteration.

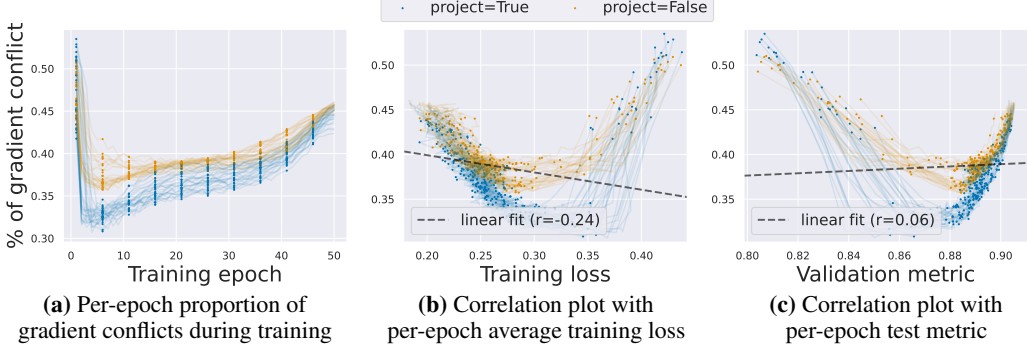

(a) Per-epoch proportion of gradient conflicts during training

(b) Correlation plot with per-epoch average training loss

(c) Correlation plot with per-epoch test metric

Figure 4: Proportion of conflicting gradients during uniform scalarization training with and without PCGrad on the 40 attribute classification tasks of CelebA while varying learning rate ([5e-4, 5e-3, 1e-2]), model depth ([3, 6, 9]) and width ([0.5, 0.75, 1]). While the overall number of encountered conflicts differs, the trend is consistent across both settings in that the higher number of conflicts encountered towards the end of training does not harm training or final model performance.

| % conflicting gradients variance ($\times 10^{-4}$) | DomainNet | Taskonomy |
|---|---|---|
| Batch size | 2.40 | n/a |
| Learning rate | 2.25 | 47.6 |
| Model width | 0.40 | 1.73 |
| Encoder depth | 1.09 | 2.63 |
| Decoder depth | n/a | 5.06 |

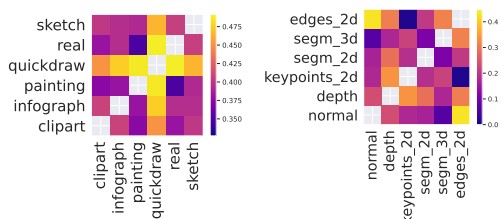

**(a)** We measure the variance across learning rates while keeping other parameters fixed (model capacity, batch size, etc.), then report the median variance across all settings. We repeat this process for each axis of variation.

**(b)** Proportion of pairwise task conflicts (for a fixed model size and learning rate) across training epochs (median), for DomainNet (6 domains) and Taskonomy (subset of 6 tasks)

Figure 5: (*left*) Variance of observed gradient conflicts when sweeping over different training hyperparameters (high impact) as well as model capacity (low impact), and (*right*) illustrating the median proportion of pairwise gradient conflicts as a measure of task affinity

Secondly, we analyze the effect of different factors of training variations on the observed proportion of gradient conflicts in Figure 5(a). On the one hand, hyperparameters directly related to weights updates, such as the learning rate of batch size, have a clear impact. In particular, since memory consumption is often a bottleneck of gradient-based MTO methods, this means that naively decreasing batch size to reduce memory usage may severely impact such methods in practice. On the other hand, model capacity has a lesser effect on the pattern of gradient conflicts, despite influencing task interference as highlighted in Figure 2. Nevertheless, we also observe that the overall magnitude of conflicts encountered for different pairs of tasks reveal interesting task affinity patterns, as illustrated in Figure 5(b): For instance, the `quickdraw` domain of DomainNet appears as a clear outlier (across all model capacities), which is also the case in terms of MDL performance in Figure 2.

In summary, we observe intriguing properties of gradient conflicts in practice, in particular suggesting that the extra cost of measuring, storing and correcting conflicting gradients at every training iteration can be superfluous. As an alternative, [30] shows that less costly regularization methods (e.g., early stopping) can be competitive with MTO. In this work, we consider an orthogonal direction, by tuning the scalarization/resampling weights hyperparameters to regulate the training speed of the different tasks/domains. However, the computational cost of browsing this large search space becomes a practical caveat as $T$ grows. In the next section, we leverage a scalable hyperparameter tuning algorithm to tackle this problem and apply it on the DomainNet and CelebA datasets.

## 5 Population-based training for scalarization weights selection

Tuning the weights **p** with classical parameter search methods, such as grid search or random search, becomes extremely costly when $T$ increases. To address this high computational demand, we propose to leverage the population-based training (PBT) [24] framework which has been used for efficient hyperparameter search in reinforcement learning [24] and for data augmentation pipelines [23].

**Population-based training (PBT).** PBT is an evolutionary algorithm for hyperparameter search: $N$ models are trained in parallel with different starting hyperparameters. Every $E_{ready}$ epochs, the models synchronize: The $Q\%$ worst models in the population are stopped, and their model weights and hyperparameters are replaced by the ones of the $Q\%$ best models (*exploit step*); Then, the newly copied hyperparameters are randomly perturbed to reach a new part of the hyperparameter search space (*explore step*). Finally, training resumes until $E$ epochs are reached. In other words, PBT enables dynamic exploration of the hyperparameter search space with a fixed computational cost (cost of training $N$ models + potential overhead from synchronization). A follow-up work, Population-based Bandits (PB2) [46] proposes to leverage Bayesian optimisation [17] to better guide the explore step. In contrast to PBT, PB2 also offers theoretical guarantees on the convergence rate.

**Using PBT to tune scalarization weights.** PBT tuning relies on three important characteristics that may conflict with the standard scalarization MDL/MTL training pipeline:

- Models are trained with a dynamic schedule of hyperparameters. While this contrasts with standard scalarization in which **p** is fixed, we do not expect this to be an issue as recent work has shown that scalarization with dynamic random weights performs well in practice [35].

- Secondly, models are compared against one another after a few epochs of training ($E_{ready}$ epochs) in contrast to e.g. random search where models are usually trained until convergence, or reaching a certain stopping criterion. Consequently, tuning $E_{ready}$ can significantly impact the search's stability and outcome, which we further discuss in Appendix 5.
- Finally, models in the population are compared using a single objective (e.g. average task/domain metric) during training. This may be an issue if the task metrics' have widely different ranges and the PBT scheduler may simply learn to favor short term improvement by giving higher weight to tasks with high metrics; While we do not observe this issue in our settings, recent work [15] also proposes a multi-objective variant of PBT which may be better fitted for MDL/MTL applications.

Nevertheless, the key advantage of PBT is its computational efficiency with regard to search space exploration: For a constant cost of training $N$ models, and some minor overhead related to checkpointing, PBT explores up to $N(1 + Q \times E_{total}/E_{ready})$ possible hyperparameter configurations throughout training of the population.

Table 1: Results of MDL when jointly training all 6 domains of DomainNet for scalarization (uniform and PBT-found weights) and MTO methods. PBT is run with a population size of $N = 12$ models, such that every $E_{ready} = 5$ epochs, $Q = 25\%$ of the population triggers an exploit/explore step.

| DomainNet (ResNet26 with 0.25 width) | | | | | | | |
|---|---|---|---|---|---|---|---|
| | **average** | clipart | infograph | painting | quickdraw | real | sketch |
| Scalarization | | | | | | | |
| Uniform | 46.78 ± 0.10 | 56.31 ± 0.04 | **20.46 ± 0.15** | 40.95 ± 0.45 | 60.69 ± 0.07 | 55.64 ± 0.01 | **46.64 ± 0.32** |
| PBT | **48.01 ± 0.08** | 58.31 ± 0.15 | 19.45 ± 0.08 | **41.32 ± 0.11** | **63.58 ± 0.31** | **60.43 ± 0.05** | 45.00 ± 0.26 |
| MTO - Loss-based | | | | | | | |
| Uncertainty [26] | 45.12 ± 0.07 | **59.24 ± 0.09** | 17.14 ± 0.25 | 37.75 ± 0.16 | 59.85 ± 0.16 | 52.35 ± 0.20 | 44.42 ± 0.10 |
| IMTL-L [38] | 44.22 ± 0.10 | 58.05 ± 0.22 | 16.41 ± 0.16 | 37.53 ± 0.21 | 59.22 ± 0.29 | 51.27 ± 0.19 | 42.83 ± 0.38 |
| MTO - Gradient-based | | | | | | | |
| CAGrad [37] | 42.82 ± 0.06 | 54.08 ± 0.03 | 18.26 ± 0.04 | 36.79 ± 0.14 | 56.46 ± 0.28 | 49.52 ± 0.15 | 41.78 ± 0.01 |
| GradDrop [8] | 42.52 ± 0.05 | 53.34 ± 0.03 | 18.16 ± 0.07 | 37.25 ± 0.02 | 55.09 ± 0.13 | 49.94 ± 0.26 | 41.35 ± 0.09 |
| PCGrad [64] | 42.78 ± 0.14 | 53.55 ± 0.04 | 18.29 ± 0.34 | 37.31 ± 0.38 | 55.60 ± 0.08 | 50.41 ± 0.13 | 41.52 ± 0.67 |

| DomainNet (ResNet26 with original width) | | | | | | | |
|---|---|---|---|---|---|---|---|
| | **average** | clipart | infograph | painting | quickdraw | real | sketch |
| Scalarization | | | | | | | |
| Uniform | 48.83 ± 0.08 | 58.69 ± 0.05 | **21.58 ± 0.39** | 42.83 ± 0.03 | 62.51 ± 0.21 | 58.31 ± 0.06 | 49.05 ± 0.15 |
| PBT | **49.27 ± 0.12** | 58.41 ± 0.50 | 19.30 ± 0.19 | **44.53 ± 0.16** | **63.08 ± 0.42** | **60.08 ± 0.16** | **50.23 ± 0.01** |
| MTO - Loss-based | | | | | | | |
| Uncertainty [26] | 46.96 ± 0.10 | **60.71 ± 0.36** | 18.74 ± 0.03 | 40.22 ± 0.40 | 60.92 ± 0.04 | 54.37 ± 0.12 | 46.79 ± 0.23 |
| IMTL-L [38] | 46.04 ± 0.21 | 59.76 ± 0.78 | 18.21 ± 0.15 | 39.12 ± 0.76 | 60.24 ± 0.39 | 53.06 ± 0.38 | 45.87 ± 0.29 |
| MTO - Gradient-based | | | | | | | |
| CAGrad [37] | 44.91 ± 0.18 | 56.56 ± 0.38 | 19.63 ± 0.32 | 38.84 ± 0.47 | 58.06 ± 0.41 | 51.80 ± 0.58 | 44.58 ± 0.45 |
| GradDrop [8] | 45.15 ± 0.08 | 56.22 ± 0.43 | 19.89 ± 0.16 | 39.70 ± 0.03 | 57.55 ± 0.12 | 52.81 ± 0.04 | 44.76 ± 0.18 |
| PCGrad [64] | 44.96 ± 0.14 | 55.79 ± 0.24 | 19.82 ± 0.20 | 39.65 ± 0.29 | 57.30 ± 0.30 | 52.57 ± 0.11 | 44.65 ± 0.65 |

In Table 1 and Table 2, we report results for searching optimal scalarization weights $\mathbf{p}^*$ when training for all 6 domains of DomainNet and for 8 tasks (attribute subsets) of CelebA. For PBT results, we first run the search algorithm using the implementation from Raytune [33]. We use 70% of the training set for training, and use the remaining 30% to rank models in the population by measuring their average accuracy on this set. Once the search is done, we retrain a model on the full training set using the scalarization weights found by PBT. For comparison, we also report results for uniform scalarization and MTO methods, using the implementation from [31]. All final models are trained for three different learning rates and the best metric is reported, averaged across 2 random seeds.

On the DomainNet example, we observe that the scalarization weights found by PBT outperforms all methods, confirming our insights that tuning weights $\mathbf{p}$ can further enhance scalarization. Note that MTO methods were not designed or employed for multi-domain settings such as DomainNet, which may explain why gradient-based MTO methods all underperform their loss-based counterparts in the results of Table 1, while they do exhibit good performance on CelebA MTL. In fact, on CelebA, while PB2 search reaches the highest overall average metric, we find that results across tasks exhibit more variance, with PB2 and CAGrad yielding the best, comparable, performance. Overall, the very narrow differences in accuracy makes it difficult to highlight one specific method, which also raises the issue on whether CelebA is a robust enough MTL benchmark. Nevertheless, both experiments

show that scalarization can outperform more complex optimization methods when its weights are tuned properly, which can be done efficiently using scalable hyperparameter search methods like PBT or PB2. We discuss further insights and compute details in Appendix 5.

Table 2: Results of MTL when training on all 8 tasks (subset of attributes) of CelebA defined in Appendix 1. For PBT and PB2 we use slightly different parameters than DomainNet to account for the fact that CelebA contains more tasks, and hence has a larger search space: All PBT runs use a population size of $N = 12$ models, such that every $E_{ready} = 3$ epochs, $Q = 40\%$ of the population triggers an exploit/explore step. For PB2 runs we use a population size of $N = 8$ and otherwise the same $Q$ and $E_{ready}$ hyperparameters. For the sake of space, we omit standard deviations in CelebA in the main text (in the range $1e^{-4}$), and only report results for the four best performing MTO baselines.

| ViT-S/4, 6 layers, full width | | | | | | | | |
| | **average** | age | clothes | face structure | facial hair | gender | hair color | hair style | mouth |
|---|---|---|---|---|---|---|---|---|---|
| Scalarization | | | | | | | | | |
| Uniform | 91.23 | 86.70 | 92.79 | 85.16 | 95.45 | 98.10 | **92.99** | 91.68 | **86.95** |
| PBT | 91.21 | 86.79 | 92.77 | 85.18 | 95.42 | 98.02 | 92.92 | 91.66 | **86.95** |
| PB2 | **91.28** | 86.96 | 92.87 | 85.17 | 95.47 | 98.10 | 92.90 | **91.81** | 86.94 |
| MTO | | | | | | | | | |
| IMTL-L [38] | 91.19 | 86.60 | 92.80 | 85.18 | 95.46 | 97.97 | 92.93 | 91.66 | 86.94 |
| CAGrad [37] | 91.27 | 86.92 | **92.89** | 85.11 | **95.49** | **98.17** | 92.93 | 91.74 | 86.89 |
| GradDrop [8] | 91.27 | **87.05** | 92.73 | **85.27** | 95.48 | 98.11 | 92.93 | 91.64 | 86.94 |
| PCGrad [64] | 91.18 | 86.71 | 92.74 | 85.09 | 95.37 | 98.12 | 92.90 | 91.66 | 86.84 |

| ViT-S/4, 9 layers, full width | | | | | | | | |
| | **average** | age | clothes | face structure | facial hair | gender | hair color | hair style | mouth |
|---|---|---|---|---|---|---|---|---|---|
| Scalarization | | | | | | | | | |
| Uniform | 91.17 | 87.33 | 92.50 | 85.10 | 95.45 | **97.93** | 92.85 | 91.39 | 86.80 |
| PBT | 91.15 | 87.43 | 92.51 | **85.18** | 95.46 | 97.78 | 92.51 | 91.36 | **87.00** |
| PB2 | **91.25** | 86.83 | **92.85** | 85.12 | **95.49** | 98.19 | 92.90 | **91.78** | 86.81 |
| MTO | | | | | | | | | |
| IMTL-L [38] | 91.16 | 87.40 | 92.44 | 85.09 | 95.44 | 97.92 | 92.87 | 91.39 | 86.75 |
| CAGrad [37] | 91.22 | 87.37 | 92.66 | 85.13 | 95.40 | 97.92 | **92.92** | 91.61 | 86.74 |
| GradDrop [8] | 91.07 | 87.41 | 92.36 | 85.01 | 95.41 | 97.73 | 92.75 | 91.21 | 86.70 |
| PCGrad [64] | 91.14 | **87.53** | 92.42 | 85.00 | 95.38 | 97.87 | 92.84 | 91.37 | 86.68 |

## 6 Limitations

Since we are building off linear scalarization, our study suffers from the same issue: It has been shown, for instance in [5], that scalarization only finds solutions on the convex parts of the Pareto front. Same as previous studies [63, 52], we do not observe this to be an issue in practice, but there may be some cases where scalarization can simply not perform as well as more advanced MTO methods. In addition, our results and observations relies on an experimental study. While we do attempt to experiment over a diverse set of benchmarks and model capacities to get as many data points as possible, we can not guarantee the generality of our claims across all MDL/MTL settings.

## 7 Conclusions

This work presents a comprehensive evaluation of scalarization's effectiveness in multi-domain and multi-task learning, spanning diverse model capacities and dataset sizes. Our analysis reveals that larger-capacity models often benefit more from joint learning across diverse settings. In addition, tuning scalarization weights is key to reach optimal performance at inference and improve the MTL/MDL model generalization. Nevertheless,, given a specific set of specific tasks/domains, the optimal weights are rather robust to changes in model capacities within the same architecture family. We then investigate the impact of model capacity on gradient conflicts observed during training and observe low correlation with MTL/MDL performance. Finally, to tackle the large search space of tuning scalarization weights, we propose to leverage population-based training as a scalable, efficient method for tuning scalarization weights as the number of tasks/domains increases.

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

# Supplemental Material to "Scalarization for Multi-Task and Multi-Domain Learning at Scale"

## 1 Experimental settings and training hyperparameters

### 1.1 Multi-Domain

We refer to benchmarks as "multi-domain" when they contain multiple input visual domains with a shared set of output classes (i.e., $\forall i \neq j, X_i \neq X_j$ and $Y_i = Y_j$).

**CIFAR-10 and STL-10.** CIFAR-10 [29] is a classical benchmark for image classification containing 50k training samples uniformly distributed across 10 classes. STL-10 [9] is a semi-supervised dataset which was designed to resemble CIFAR-10. Specifically, we only use the 5000 annotated images in STL-10, which are also uniformly distributed across the same 10 classes as CIFAR. In STL-10, the images themselves are from the ImageNet [54] dataset, and cropped/resized to 96 pixels. We further resize them to 32 pixels to align with CIFAR. In summary, the key difficulties are (i) the input distribution shift between the two datasets and (ii) the high imbalance in data availability.

We use a vision transformer backbone (ViT-S) optimized for small-scale datasets [18] (compared to the original ViT-S, this backbone contains smaller patch sizes, fewer transformer layers and narrower embeddings, but a higher number of heads). To control model capacity, we vary the depth (number of transformer layers) in $\{3, 6, 9\}$ and the width (token dimension) in $\{48, 96, 144, 192\}$, Finally, we train each model from scratch on a single NvidiaV100 GPU with a batch size of 256 images for 300 epochs (including 30 epochs of linear learning rate warmup), using a learning rate of $0.001$ and weight decay of $0.05$ with the AdamW optimizer and cosine learning rate decay.

**DomainNet.** DomainNet [47] is a classification dataset of 6 visual domains annotated for 345 classes, for a total of roughly 410k training samples. DomainNet was initially introduced for the problem of multi-source domain adaption, in which one or more of the domains does not have training annotations; the key difficulty is thus to learn representations that are aligned across domains. In contrast to the CIFAR+STL example, DomainNet exhibits distribution shifts across both the input domains and output classes, as visualized in Figure A.

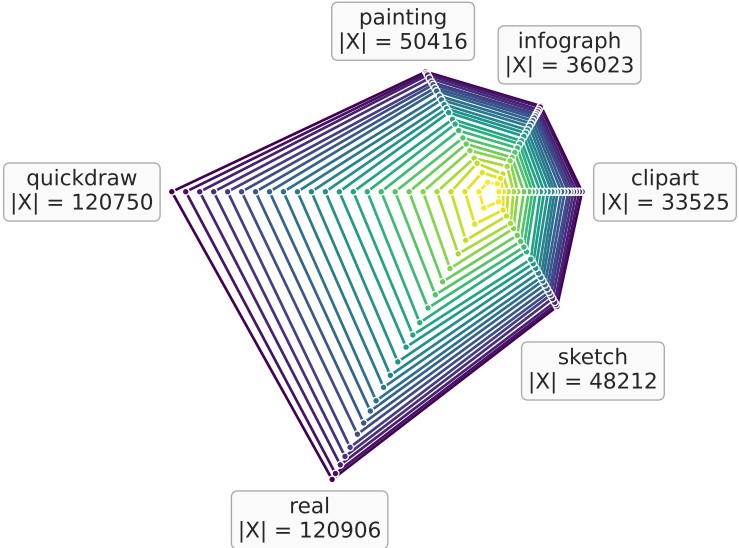

Figure A: Illustrating the data imbalance in DomainNet with a contour plot of the number of samples per class and domains in DomainNet. Each of the corners of the hexagon represents one of the six domains in DomainNet, and the lines (levels of the contour plot) represent the number of samples, drawn every 15 classes. For comparison, a uniformly distributed dataset would yield perfect hexagons.

Following previous literature [47, 32], we use a ResNet-101 as our main backbone. There is no domain-dependent layer in the architecture: the final classifier layer is shared across all domains. We first perform a training sweep over the largest domain (`real`) to select the best-performing learning rate from $\{0.3, 0.03, 0.003, 0.0003\}$ and the number of epochs from $\{30, 60, 90\}$. Following these results, we use a learning rate of $0.03$ and train for 30 epochs with a batch size of 512 in subsequent experiments. We train with the AdamW optimizer with a weight decay of $1e-4$. We also apply linear learning rate warm-up during the first five training epochs and use cosine schedule learning rate decay for the rest of the training. Finally, to control model capacity, we vary the depth (backbone) in {ResNet-26, ResNet-50, ResNet-101} and the width factor.

**Multi-domain, resampling and training length**    In the multi-domain setting, scalarization weights become resampling probabilities for each dataset, as shown in (2). Consequently, the notion of "epoch" is hard to define compare to the standard mono-dataset setting. To resolve this, we always define epochs with respect to one of the domains. For instance, in the CIFAR+STL case, we use STL as our reference. Therefore, "one epoch" translates to seeing as many samples as in the original STL dataset (5000) using the current batch size (256), i.e. roughly 20 training steps. In the DomainNet case, we define epochs relatively to the `real` domain. This definition has the advantage of not being impacted by the sampling weights **p**; In particular, this means that both the MDL models and the single dataset (SD) baselines are trained for as many training steps, and see the same amount of training samples, only sampled from different data distributions.

## 1.2   Multi-task

We define multi-task benchmarks as datasets where every image is fully annotated for multiple output tasks (i.e., $\forall i \neq j, X_i = X_j$ and $Y_i \neq Y_j$). This setting is particularly popular for scene understanding problems where every scene is labelled with multiple dense predictions (e.g. depth, normals, segmentation mas, edges, etc.)

**Celeba.**    CelebA is a binary attribute classification dataset containing 40 attributes and roughly 162k training images. To turn CelebA into a multi-task problem, it is common to consider each attribute as a binary classification task: More specifically, we use a fully shared backbone with a final linear layer of 40 outputs, outputting logits for every task. The model is then trained using 40 binary cross-entropy losses, one for each attribute. To make our comparative analysis more scalable, we define several tasks as subsets of attributes, grouped based on semantic similarity (e.g. all hair colors are in the same subgroup). The 8 resulting subsets of attributes are described in Table A. In the scalarization setting, this simply means that some of the attributes share the same importance weight.

As a backbone, we use the same ViT-S/4 based architecture as for CIFAR/STL. We train for 50 epochs with 5 epochs of learning rate warmup. We use a learning rate of $0.0005$ with cosine schedule decay anf train with the AdamW optimizer with a weight decay of $0.05$. We use input images of size 32 (with tokens of size 4), a batch size of 256, and RandAugment data augmentations.

Table A: The eight tasks defined as subsets from CelebA attributes used in our main analysis. Attributes in the same subset share a common importance weight $p$

| Hair color | Hairstyle | Facial Hair | Mouth | Clothes | Face Structure | Gender | Age |
|---|---|---|---|---|---|---|---|
| Black Hair | Bald | 5'o'Clock Shadow | Big Lips | Eyeglasses | Big Nose | Male | Young |
| Blond Hair | Bangs | Mustache | Mouth Slightly Open | Heavy Makeup | Chubby | | |
| Brown Hair | Receding Hairline | No Beard | Smiling | Wearing Earrings | Double Chin | | |
| Gray Hair | Sideburns | Goatee | Wearing Lipstick | Wearing Hat | High Cheekbones | | |
| | Straight Hair | | | Wearing Necklace | Oval Face | | |
| | Wavy Hair | | | Wearing Necktie | Pointy Noise | | |

**Taskonomy.**    Taskonomy [65] is a large dataset containing a variety of dense prediction tasks for indoor scenes. We use the `tiny` split of Taskonomy which contains roughly 275k images. Taskonomy was originally introduced for the problem of task clustering: The original work [65] proposes a task affinity metric to define a taxonomy of tasks. This taxonomy structure is then used to determine which tasks should be trained from scratch and which tasks could benefit from others via transfer learning. Closer to our setting, follow-up works [58, 16] propose to use this taxonomy to determine which tasks should be grouped or not in multi-task learning. Once the groupings are determined, a

separate backbone is trained for each group of tasks. Instead, for our analysis, we use Taskonomy-tiny in a more standard multi-task framework, where a backbone is shared across tasks.

For training, we follow the methodology of [58]. We use a ResNet-26 backbone (with varying bottleneck width) with a mirrored decoder; By default, only the encoder is shared across tasks and each task receives its own decoder. To vary model capacity, we add the option to share more or fewer layers of the decoders across tasks. We use the same learning rate of 0.1 and training for 100 epochs using a batch size of 256. We train with SGD with a momentum of 0.9 and a weight decay of $1e - 4$. Following [58], all output prediction maps are rescaled to have zero mean and unit variance on the training set, and all dense tasks are trained with L1 loss.

## 2    Additional analysis results

### 2.1    Complete results and methodology for Figure 2

In Section 4, we perform MDL and MTL experiments on several pairs of datasets, each time comparing to the single dataset (SD) baseline trained for the same model capacity and training length. All results are run for three random seeds on CelebA and CIFAR+STL, and two random seeds for DomainNet and Taskonomy. To present these results in a condensed form in Figure 2, we first find the scalarization weights $\mathbf{p}^* = (p_1^*, 1 - p_1^*)$ that yield the best average accuracy across both datasets. Then we report the difference in metrics between MDL trained with weights $\mathbf{p}^*$ and the corresponding SD baseline, for each dataset. Note that for the Taskonomy case, where the tasks are evaluated via L1 loss, we measure the negative difference instead to keep the same interpretation as the other settings where a positive value means MDL improves over SD.

For completeness, we report all results for the CIFAR/STL case as trade-off plots (accuracy on dataset 1 versus accuracy on dataset 2) in Figure B (CIFAR/STL) and in in Figure C (segmentation 2D and depth tasks of Taskonomy). We observe the same trends as summarized in the main paper: First, when increasing model capacity, the MDL performance over the SD baseline increases; This is best seen when width increases (across columns). Second, the optimal weights vary across model sizes: At low width, the best performance is reached for a ratio in the range of $[0.3, 0.4]$. While larger models prefer $p_{\text{STL}}^* \in [0.1, 0.3]$. Finally, it is interesting to note that these weights also differ from heuristics commonly used to set scalarization weights: such as uniform scalarization $p_{\text{STL}} = 0.5$ or for instance setting the weights to match the number of samples in each dataset $p_{\text{STL}} = 0.09$. This further highlights the fact that tuning scalarization weights can make scalarization into a stronger baseline for MDL/MTL.

### 2.2    MDL helps generalization

When comparing the MDL/MTL and SD performance, we often observe that MDL/MTL improvements over SD are visible at inference but not at training time (as illustrated for instance in Figure D). This indicates that MDL/MTL training helps generalization of the model compared to training on a single dataset. In particular, in the MDL setting, this draws an interesting parallel with data augmentations: In fact, MDL training can be seen as adding additional input data from a new distribution, with a probability given by the scalarization weights $\mathbf{p}$, while sharing the same semantic classes. And similarly to data augmentations, adding this extra data source makes the training distribution harder to fit (hence the SD baseline outperforming MDL at train time) but can greatly benefit generalization performance (hence the inverse trend at inference).

To push the analogy further, the experimental study of [41] suggests that a good data augmentation should be one with a high *affinity* to the original data distribution (i.e., the distribution shift between the original data and augmented one should not be too significant) as well as a high *diversity* (i.e., the added data should be complex enough, which can be measure e.g. with magnitude of the training loss). This hypothesis also matches our observations: For instance adding `infograph` data to the `real` domain leads to a low affinity pairing but with high diversity and yields weaker MDL performance on the `real` images compared to the SD baseline (see Figure 2).

Finally, we note that the problem of finding optimal scalarization weights mirrors the one of finding data augmentation hyperparameters, which has been extensively explored for the task of image classification. It has led to now commonly-adopted augmentation strategies such as RandAugment [12], AutoAugment [11], or PBA [23], which also uses Population-based training to tackle this problem.

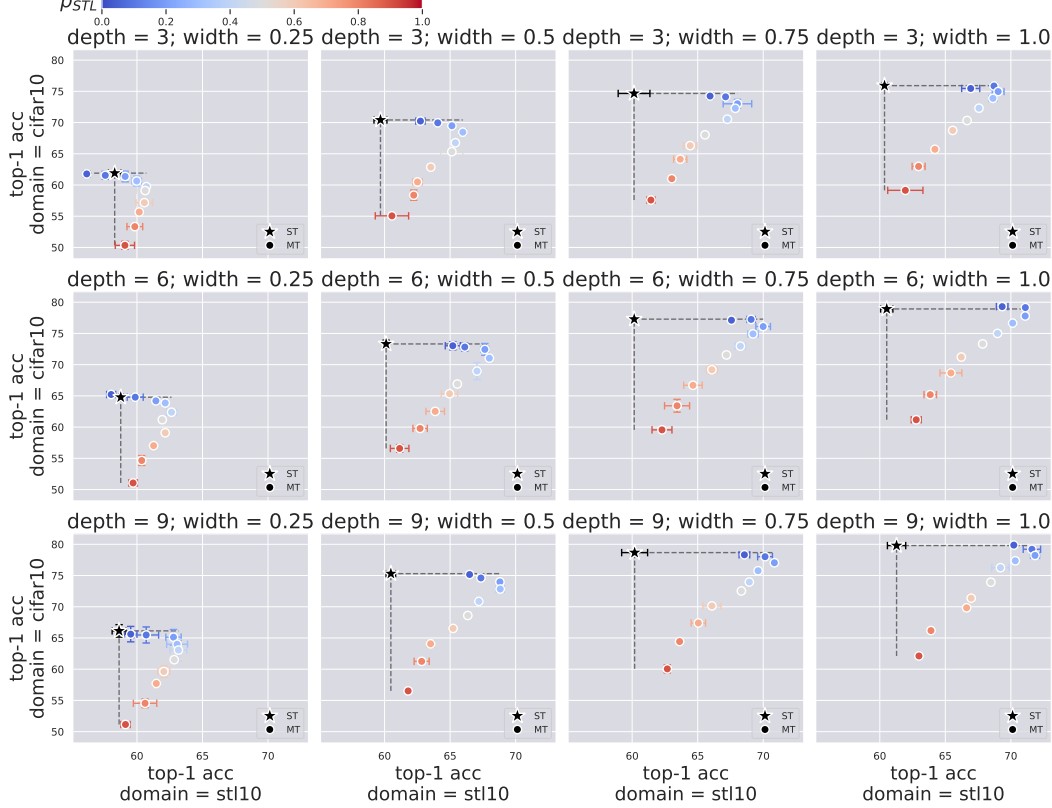

Figure B: Complete analysis results for the CIFAR+STL scenario. Each row corresponds to a different model depth and each column to a model width, in increasing order. In each plot, we plot the model's test accuracy on CIFAR-10 versus the test accuracy on STL-10. The single dataset baseline (SD) is drawn in black and corresponds to the accuracy obtained when training two separate networks, one on each dataset independently. The circle markers correspond to the MDL model trained for different scalarization weights $\mathbf{p}$. The value of $p_{STL}$ is represented as the color of the marker, while the remaining weight is always set to $p_{CIFAR} = 1 - p_{STL}$.

# 3 Methodology for measuring gradients conflicts

In this appendix, we briefly describe our methodology for Section 4.2. We use the same definition of gradient conflicts as in PCGrad [64]: Two task/domain specific gradients are conflicting if and only if the cosine of the angle between them is strictly negative. We train a model using standard uniform scalarization, measure the number of conflicting pairs of task/domain gradients over one epoch of training, and report it as a percentage (of all pairs), for each epoch during training.

We report these results in the main text in Figure 4 and in Figure E below. Our main observation is that the presence or absence of gradient conflicts does not correlate well with actual MDL/MTL performance throughout trainined. This challenges the assumption underlying many multi-task optimization (MTO) methods [64, 38, 62] that reducing gradient conflicts leads to improved MTL performance. This also aligns well with recent results of [63, 30] showing that MTO methods that reduce gradient conflicts do not outperform simpler scalarization approaches in practice, and with the hypothesis of [30] that many MTO methods can be reinterpreted as regularization techniques.

# 4 Consistency of optimal scalarization weights

As noted in the analysis from Section 4, the optimal weights $\mathbf{p}^*$ is rather consistent across model depths and widths. For instance, on the CIFAR/STL case, $\mathbf{p}^*$ always falls in the range of [0.2, 0.4].

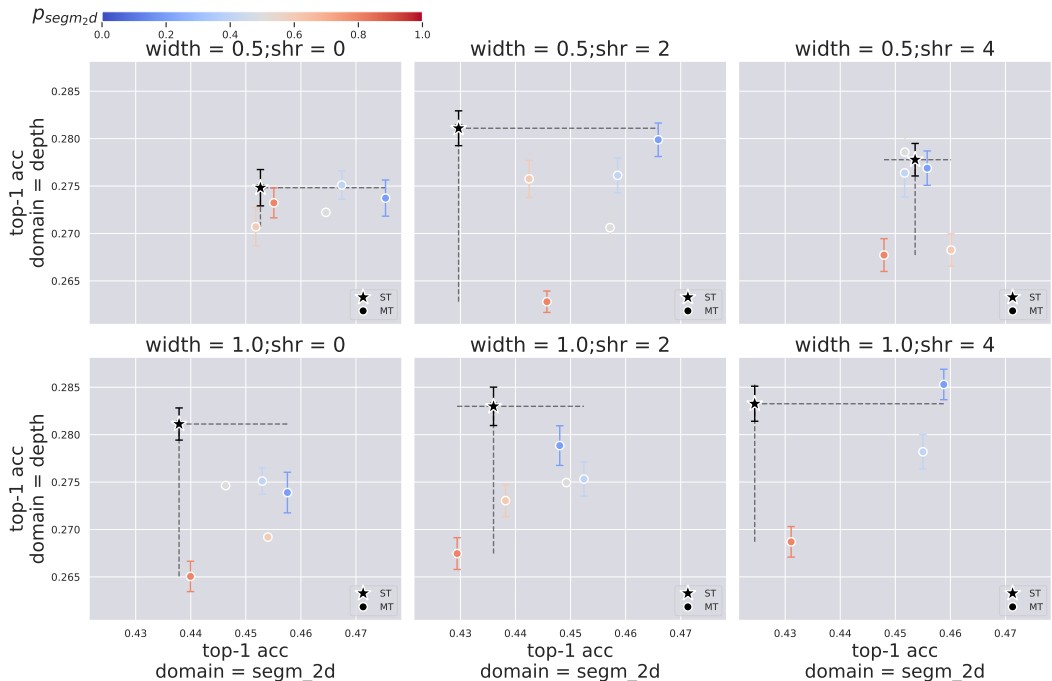

Figure C: Complete analysis results for the segmentation 2D and depth prediction tasks in the Taskonomy scenario. Each row corresponds to a different number of layers shared in the decoder (`shr`) and each column to a model width. In each plot, we plot the model's test Le loss of each task on either axis. The single dataset baseline (SD) is drawn in black and corresponds to the accuracy obtained when training two separate networks, one on each dataset independently. The circle markers correspond to the MDL model trained for different scalarization weights $\mathbf{p}$. The value of $p_{\text{segmentation2D}}$ is represented as the color of the marker.

This can also be seen from the qualitative results of the population-based training search of scalarization weights in Section 5.2: While the history of hyperparameter changes during the search differ, PBT tends to converge to similar distribution for the scalarization weights $\mathbf{p}$ across different model depths and widths. This suggests that the theoretical search space for $\mathbf{p}$ may be reduced in practice leading to a more computationally efficient search: Performing a rough initial search on a smaller model from the same architecture family can provide a promising range for $\mathbf{p}$, and can then be refined by searching with the larger target architecture.

## 5 PBT results

Because the search space for scalarization weights $\mathbf{p}$ grows exponentially with the number of tasks, classical hyperparameter search methods such as grid search or random search would struggle to scale as the number of tasks increases. Bayesian optimization (BO) [17] allows for faster results by browsing the search space in a smart way by building and following a probabilistic model of the hyperparameters. However, BO still requires training models to convergence (or until an early stopping criterion is met) which can be computationally expensive. Instead, we experiment with the Population-based Training framework [24] for searching for the optimal scalarization weights. PBT relies on the assumption that the "goodness" of a certain hyperparameter choice can be evaluated in a few epochs during training, rather than having to finish a full run of training.

### 5.1 Compute resources

As mentioned in the main paper we perform all training runs on a single NVIDIA V100 machine with 32GB of memory. However, both the PBT and MTO models require higher compute resources than a single normal run of training, which we discuss below.

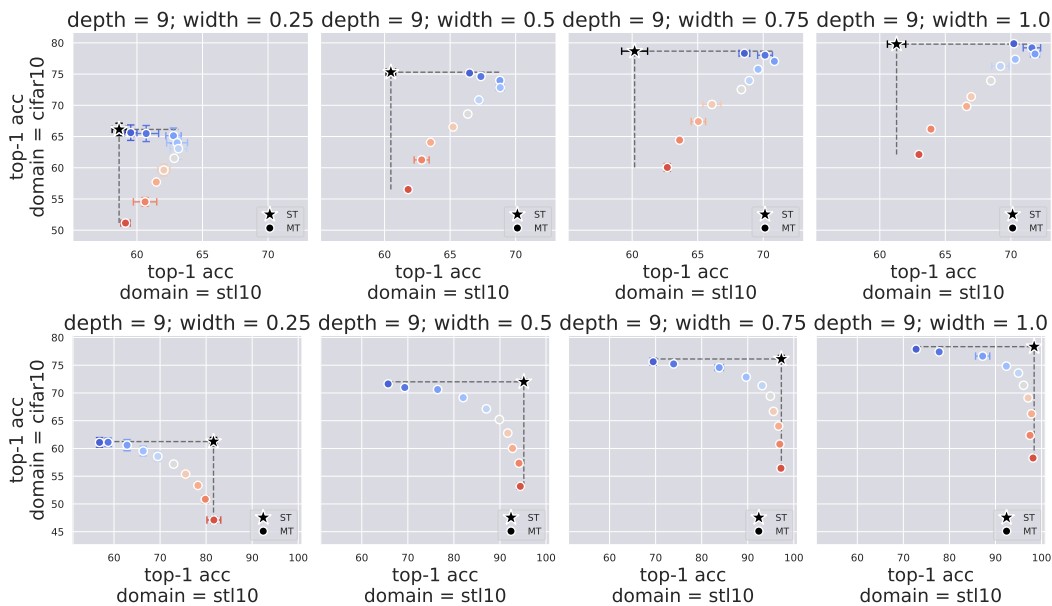

Figure D: Train-test Discrepancy when comparing MDL/MTL improvement over the SD baseline visualized on the CIFAR+STL example. In particular, in the MDL setting, this matches the classical interpretation of data augmentations: Adding additional semantically relevant data from an input distribution may be harder to fit at training time but leads to improved generalization performance at inference.

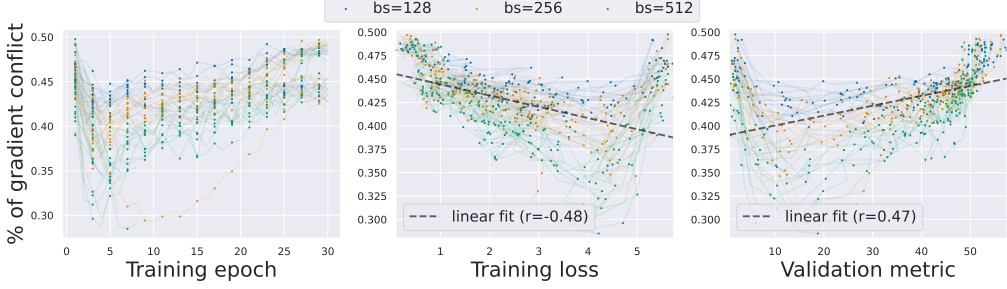

(a) DomainNet. Individual curves are colored by batch size.

Figure E: Additional result on measuring conflicting gradients when training with standard uniform scalarization for DomainNet (6 domains).

- **PBT** requires training a population of **N** models that are regularly synchronized; then followed by a final training run with the found optimal weights. Following previous works [23], we perform the hyperparameter search on a smaller subset of the data (in our experiments $r = 0.7$ fraction of the training set). In summary, the expected computational cost is roughly $Nr + 1$ times higher than standard training. On the CelebA example, we also observe that using PB2 [46], which combines the benefits of Bayesian Optimisation and PBT, yields better hyperparameters using a smaller population size. In terms of memory usage, PBT is the same as a standard training run: Synchronization is handled via checkpoints saving and loading, such that only one model lives in memory. Finally, we use the publicly available PBT implementation from Raytune[33] which handles all synchronization operations across the population. The implementation would also scale well to more compute resources, as the Ray API allows for easy parallelization.

- **MTO**. As shown in Figure 1, the bottleneck in most gradient-based methods is memory usage. Consequently, this requires us to decrease the batch size to meet memory requirements and compensate with gradient accumulation (or parallelism if multiple devices are available).

For instance, on the DomainNet experiments with all 6 domains, we need to decrease from a batch size of 512 to a batch size of 128 with 4 steps of accumulation to still fit in memory requirements. This also raises the question of how to handle synchronization across batches: For instance, in PCGrad, the gradient conflicts (and projections) can be computed either **(i)** per local batch, before accumulation: this may lead to noisier updates; or **(ii)** after gradients are accumulated: However this is more memory-intensive as this requires to store the previously accumulated per-task gradient as well as the one being currently computed. In practice we use the implementation of [31] for all MTO methods.

When comparing different hyperparameter searches for scalarization, PBT allows for much faster exploration of the search space than classical techniques such as grid search. However, comparing PBT+scalarization with MTO is less straightforward as the computational cost depends on many factors (e.g. population size, number of tasks, and impact on memory usage, etc.), but generally, the "scalarization + hyperparameter search" approach is more favorable in case of low memory requirements as it does not change memory costs compared to standard training. However, soTA gradient-based methods are not very costly for a low number of tasks (e.g. 2-3) as shown in Table 1 which makes them appealing in settings with a few tasks. Nevertheless, one of our key takeaways is that allocating extra resources for tuning scalarization weights, to mirror the extra resources needed for MTO training, makes scalarization into a much stronger baseline, on-par or even outperforming MTO methods as shown in Section 5. Finally, another important difference is that hyperparameter search methods directly optimize for the target objective: the optimal hyperparameters are found by maximizing the average task/domain accuracy on a hold-out validation dataset. In contrast, MTO methods optimize for a proxy metric (such as reducing gradients conflicts) that may not always correlate with final performance as shown in Section 4.2.

## 5.2 Qualitative results

In this section, we report some qualitative results of the hyperparameter scheduled found by PBT and PB2. At the end of PBT search, we select the model with the highest validation performance and backtrack its history to backtrack its choices of hyperparameter values during training: This yields the policy of optimal weights found by PBT which is then used to retrain a model on the full training set. We also experimented with retraining a model using the last weights of the policy, but this usually slightly underperform using the whole history of weights in the majority of cases.

In Figure F, we report examples of the policy of weights found by PBT and PB2 search for the parameters $E_{ready} = 3$, $Q = 40\%$ and $N = 12$ for PBT (respectively $N = 8$ for PB2).

## 5.3 Quantitative results

Here we report additional results for the CelebA and DomainNet experiments of Section 5, in the style of Table 1 and Table 2 in the main text: We include results using a ResNet-50 on DomainNet in Table B, and results for additional widths in CelebA in Table C and Table D.

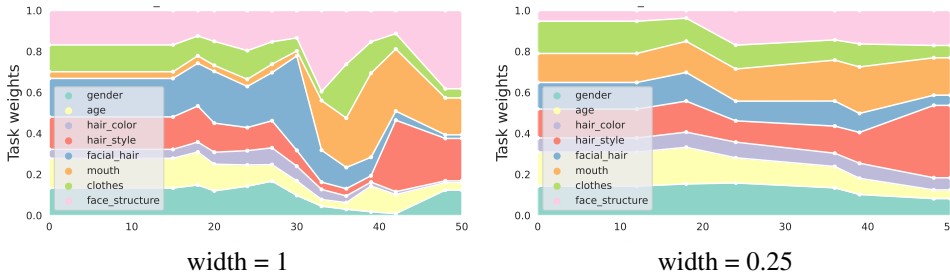

**(a)** Comparing Population-based training results for a depth of 9 layers with full width (left) and a smaller model with a quarter of the width (right). While both policy are quite different across training epochs, they converge towards similar distribution: For instance the weights for tasks "hair style", "gender" and "age" are significantly smaller than the one for the "mouth" and "hair style" tasks.

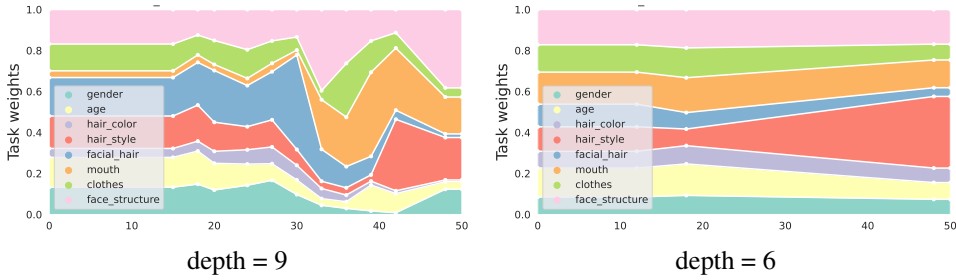

**(b)** Comparing Population-based training results for a depth of 9 layers with full width (left) and a smaller model with a depth of 6 layers (right). Similarly to the results on varying width in **(a)**, both search converge to similar distribution in task weights

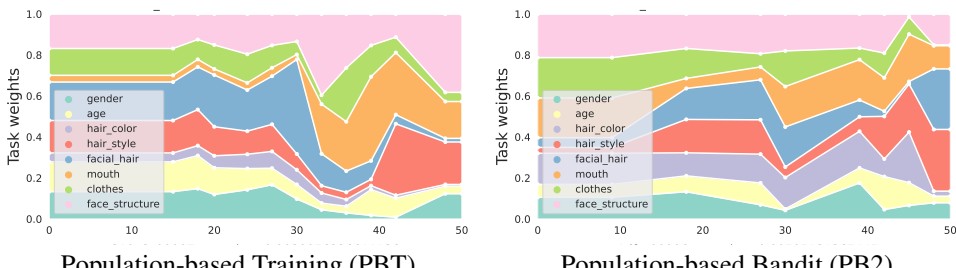

**(c)** Comparing Population-based training results for a depth of 9 layers with full width (left) and the same search with Population-based bandit (right). The two search algorithms converge to significantly different results in particular regarding weights for the "mouth" and "facial hair" tasks. This suggests that (i) there may be multiple good local minima in the search space of **p** and (ii) the heuristic used in the explore step has a significant impact on how the resulting policy.

Figure F: Qualitative results for Population-based training search on CelebA. The x-axis represents training epochs. The y-axis represents the policy scalarization weights for each task as a cumulative histogram for the run of the population with highest validation accuracy

Table B: Results of MDL when jointly training on all 6 domains of DomainNet for scalarization (uniform and PBT-found weights) and MTO methods with a **ResNet50** backbone. PBT is run with a population size of $N = 12$ models, such that every $E_{ready} = 5$ epochs, $Q = 25\%$ of the population triggers an exploit/explore.

| DomainNet (ResNet50 with 0.25 width) | | | | | | | |
|---|---|---|---|---|---|---|---|
| | **average** | clipart | infograph | painting | quickdraw | real | sketch |
| Scalarization | | | | | | | |
| Uniform | 49.69 ± 0.05 | 59.90 ± 0.15 | 22.45 ± 0.01 | 43.90 ± 0.14 | 63.13 ± 0.15 | 58.95 ± 0.09 | 49.79 ± 0.10 |
| PBT | 50.69 ± 0.10 | 61.69 ± 0.45 | 21.27 ± 0.10 | 44.72 ± 0.27 | 63.96 ± 0.13 | 62.43 ± 0.13 | 50.06 ± 0.17 |
| MTO - Loss-based | | | | | | | |
| Uncertainty [26] | 40.51 ± 0.19 | 53.33 ± 0.67 | 15.70 ± 0.02 | 34.44 ± 0.41 | 54.27 ± 0.61 | 47.86 ± 0.39 | 37.45 ± 0.37 |
| IMTL-L [38] | 37.04 ± 0.17 | 48.85 ± 0.59 | 13.93 ± 0.42 | 30.64 ± 0.36 | 51.60 ± 0.34 | 44.12 ± 0.21 | 33.12 ± 0.50 |
| MTO - Gradient-based | | | | | | | |
| CAGrad [37] | 39.82 ± 0.10 | 50.68 ± 0.05 | 16.94 ± 0.04 | 34.37 ± 0.35 | 52.16 ± 0.47 | 46.59 ± 0.00 | 38.20 ± 0.07 |
| GradDrop [8] | 39.18 ± 0.15 | 49.80 ± 0.04 | 16.77 ± 0.65 | 33.95 ± 0.52 | 51.18 ± 0.06 | 46.04 ± 0.28 | 37.36 ± 0.20 |
| PCGrad [64] | 39.48 ± 0.31 | 50.42 ± 0.97 | 16.83 ± 0.31 | 34.63 ± 0.92 | 51.14 ± 0.53 | 46.37 ± 0.51 | 37.49 ± 0.98 |

| DomainNet (ResNet50 with original width) | | | | | | | |
|---|---|---|---|---|---|---|---|
| | **average** | clipart | infograph | painting | quickdraw | real | sketch |
| Scalarization | | | | | | | |
| Uniform | 51.53 ± 0.06 | 61.89 ± 0.12 | 23.63 ± 0.01 | 45.87 ± 0.01 | 64.50 ± 0.08 | 61.46 ± 0.09 | 51.83 ± 0.33 |
| PBT | 51.83 ± 0.06 | 62.22 ± 0.06 | 22.61 ± 0.20 | 46.61 ± 0.29 | 64.71 ± 0.05 | 61.91 ± 0.05 | 52.93 ± 0.10 |
| MTO - Loss-based | | | | | | | |
| Uncertainty [26] | 42.90 ± 0.20 | 56.24 ± 0.44 | 17.36 ± 0.12 | 36.91 ± 0.82 | 56.11 ± 0.08 | 50.33 ± 0.48 | 40.49 ± 0.51 |
| IMTL-L [38] | 39.69 ± 0.13 | 52.51 ± 0.59 | 15.51 ± 0.14 | 33.45 ± 0.14 | 53.54 ± 0.05 | 46.37 ± 0.48 | 36.75 ± 0.17 |
| MTO - Gradient-based | | | | | | | |
| CAGrad [37] | 41.90 ± 0.13 | 53.32 ± 0.41 | 17.94 ± 0.42 | 36.72 ± 0.39 | 54.15 ± 0.03 | 48.31 ± 0.25 | 40.94 ± 0.14 |
| GradDrop [8] | 42.15 ± 0.14 | 53.38 ± 0.54 | 18.68 ± 0.33 | 37.00 ± 0.45 | 53.85 ± 0.11 | 49.04 ± 0.27 | 40.95 ± 0.04 |
| PCGrad [64] | 41.94 ± 0.20 | 53.46 ± 0.69 | 18.20 ± 0.28 | 36.95 ± 0.66 | 53.29 ± 0.13 | 48.88 ± 0.48 | 40.87 ± 0.49 |

Table C: (Table best seen zoomed in PDF) Results of MTL when training on all 8 tasks (subset of attributes) of CelebA for a depth of 6 layers For PBT and PB2 we use slightly different parameters than DomainNet to account for the fact that CelebA contains more tasks, and hence has a larger search space: All PBT runs use a population size of $N = 12$ models, such that every $E_{ready} = 3$ epochs, $Q = 40\%$ of the population triggers an exploit/explore step. For PB2 runs we use a population size of $N = 8$ and otherwise the same $Q$ and $E_{ready}$ hyperparameters.

**ViT-S/4, 6 layers, 0.25 width**

| | average | age | clothes | face structure | facial hair | gender | hair color | hair style | mouth |
|---|---|---|---|---|---|---|---|---|---|
| Scalarization | | | | | | | | | |
| Uniform | 90.82 ± 2.1e-04 | 86.96 ± 8.5e-04 | 92.23 ± 6.8e-04 | 84.64 ± 6.6e-04 | 95.30 ± 1.8e-04 | 97.61 ± 7.1e-04 | 92.58 ± 3.4e-04 | 91.03 ± 5.8e-04 | 86.23 ± 4.4e-04 |
| PBT | 90.93 ± 1.8e-04 | 86.94 ± 7.6e-04 | 92.37 ± 3.3e-04 | 84.78 ± 4.5e-04 | 95.18 ± 2.1e-04 | 97.58 ± 5.3e-04 | 92.82 ± 4.3e-04 | 91.48 ± 3.5e-04 | 86.25 ± 7.0e-04 |
| PB2 | 90.90 ± 2.0e-04 | 87.10 ± 1.2e-03 | 92.13 ± 3.2e-04 | 84.82 ± 5.6e-04 | 95.32 ± 2.8e-04 | 97.41 ± 4.8e-04 | 92.67 ± 5.6e-04 | 91.23 ± 2.3e-04 | 86.50 ± 2.1e-04 |
| MTO - Loss-based | | | | | | | | | |
| Uncertainty [26] | 90.82 ± 1.9e-04 | 86.94 ± 1.1e-03 | 92.22 ± 7.2e-04 | 84.63 ± 3.0e-04 | 95.32 ± 2.7e-05 | 97.62 ± 2.8e-04 | 92.57 ± 3.9e-04 | 91.02 ± 4.1e-04 | 86.22 ± 1.9e-04 |
| IMTL-L [38] | 90.82 ± 2.0e-04 | 86.94 ± 1.2e-03 | 92.22 ± 7.3e-04 | 84.63 ± 3.1e-04 | 95.32 ± 2.7e-05 | 97.62 ± 3.2e-04 | 92.57 ± 3.8e-04 | 91.02 ± 4.1e-04 | 86.22 ± 1.6e-04 |
| MTO - Gradient-based | | | | | | | | | |
| CAGrad [37] | 90.92 ± 2.7e-04 | 86.96 ± 2.1e-03 | 92.35 ± 3.0e-05 | 84.92 ± 1.2e-04 | 95.38 ± 3.9e-04 | 97.56 ± 1.4e-04 | 92.73 ± 3.8e-04 | 91.30 ± 2.2e-04 | 86.14 ± 8.9e-05 |
| GradDrop [8] | 90.65 ± 2.9e-04 | 86.76 ± 1.7e-03 | 92.03 ± 8.1e-05 | 84.48 ± 1.2e-04 | 95.23 ± 3.9e-04 | 97.41 ± 5.7e-04 | 92.46 ± 2.1e-04 | 90.83 ± 1.4e-03 | 85.98 ± 3.5e-05 |
| PCGrad [64] | 90.86 ± 1.5e-04 | 87.04 ± 1.1e-04 | 92.32 ± 7.3e-04 | 84.65 ± 4.0e-04 | 95.27 ± 1.9e-04 | 97.62 ± 3.2e-04 | 92.64 ± 3.7e-04 | 91.16 ± 2.8e-04 | 86.22 ± 5.9e-04 |

**ViT-S/4, 6 layers, 0.5 width**

| | average | age | clothes | face structure | facial hair | gender | hair color | hair style | mouth |
|---|---|---|---|---|---|---|---|---|---|
| Scalarization | | | | | | | | | |
| Uniform | 91.28 ± 2.8e-04 | 87.33 ± 1.9e-03 | 92.71 ± 8.6e-05 | 85.15 ± 2.8e-04 | 95.49 ± 2.2e-04 | 98.03 ± 6.0e-04 | 92.99 ± 3.2e-04 | 91.63 ± 5.8e-04 | 86.87 ± 3.8e-04 |
| PBT | 91.29 ± 1.7e-04 | 87.56 ± 3.3e-04 | 92.77 ± 1.6e-04 | 85.34 ± 2.5e-04 | 95.40 ± 2.9e-04 | 97.81 ± 6.4e-04 | 92.99 ± 2.1e-04 | 91.56 ± 7.2e-04 | 86.89 ± 7.6e-04 |
| PB2 | 91.38 ± 2.3e-04 | 87.71 ± 1.1e-03 | 92.84 ± 3.3e-04 | 85.20 ± 6.0e-04 | 95.54 ± 4.5e-04 | 98.00 ± 7.3e-04 | 92.95 ± 6.2e-04 | 91.73 ± 2.2e-04 | 87.05 ± 7.0e-04 |
| MTO - Loss-based | | | | | | | | | |
| Uncertainty [26] | 91.30 ± 3.1e-04 | 87.52 ± 2.2e-03 | 92.72 ± 1.4e-04 | 85.14 ± 4.6e-04 | 95.48 ± 4.7e-04 | 98.08 ± 5.3e-04 | 92.97 ± 3.5e-04 | 91.65 ± 1.9e-04 | 86.86 ± 6.4e-04 |
| IMTL-L [38] | 91.30 ± 2.6e-04 | 87.49 ± 1.4e-03 | 92.74 ± 6.6e-05 | 85.14 ± 1.0e-04 | 95.51 ± 1.3e-04 | 98.06 ± 7.8e-04 | 92.98 ± 5.8e-04 | 91.66 ± 6.7e-04 | 86.85 ± 1.0e-03 |
| MTO - Gradient-based | | | | | | | | | |
| CAGrad [37] | 91.32 ± 3.2e-04 | 87.47 ± 2.3e-03 | 92.86 ± 4.9e-04 | 85.26 ± 3.1e-04 | 95.51 ± 4.3e-04 | 98.03 ± 0.0e+00 | 93.00 ± 5.9e-04 | 91.73 ± 1.5e-04 | 86.72 ± 2.3e-04 |
| GradDrop [8] | 91.19 ± 2.3e-04 | 87.42 ± 8.5e-04 | 92.65 ± 5.2e-04 | 85.03 ± 9.5e-04 | 95.43 ± 2.9e-04 | 97.96 ± 6.7e-04 | 92.80 ± 1.8e-04 | 91.54 ± 8.5e-04 | 86.70 ± 3.0e-04 |
| PCGrad [64] | 91.30 ± 4.6e-04 | 87.45 ± 2.8e-03 | 92.72 ± 6.0e-04 | 85.16 ± 8.9e-04 | 95.54 ± 5.0e-04 | 98.08 ± 2.1e-03 | 92.97 ± 1.6e-04 | 91.63 ± 1.2e-04 | 86.82 ± 3.8e-04 |

**ViT-S/4, 6 layers, full width**

| | average | age | clothes | face structure | facial hair | gender | hair color | hair style | mouth |
|---|---|---|---|---|---|---|---|---|---|
| Scalarization | | | | | | | | | |
| Uniform | 91.23 ± 3.6e-04 | 86.70 ± 2.7e-03 | 92.79 ± 4.5e-04 | 85.16 ± 4.8e-04 | 95.45 ± 4.8e-04 | 98.10 ± 2.1e-04 | 92.99 ± 3.6e-04 | 91.68 ± 4.7e-05 | 86.95 ± 4.8e-04 |
| PBT | 91.21 ± 2.7e-04 | 86.79 ± 1.1e-03 | 92.77 ± 5.8e-04 | 85.18 ± 3.6e-04 | 95.42 ± 8.9e-04 | 98.02 ± 4.5e-04 | 92.92 ± 6.4e-04 | 91.66 ± 1.0e-03 | 86.95 ± 6.7e-04 |
| PB2 | 91.28 ± 2.5e-04 | 86.96 ± 1.5e-03 | 92.87 ± 2.7e-04 | 85.17 ± 2.4e-04 | 95.47 ± 6.5e-05 | 98.10 ± 3.9e-04 | 92.90 ± 5.8e-04 | 91.81 ± 2.6e-04 | 86.94 ± 1.1e-03 |
| MTO - Loss-based | | | | | | | | | |
| Uncertainty [26] | 91.22 ± 2.1e-04 | 86.60 ± 1.3e-03 | 92.85 ± 6.4e-04 | 85.24 ± 1.7e-04 | 95.43 ± 8.0e-05 | 98.07 ± 3.5e-04 | 92.91 ± 7.7e-04 | 91.72 ± 2.1e-04 | 86.93 ± 4.4e-05 |
| IMTL-L [38] | 91.21 ± 2.0e-04 | 86.65 ± 7.1e-04 | 92.83 ± 5.2e-04 | 85.17 ± 1.8e-04 | 95.42 ± 5.5e-04 | 98.01 ± 6.0e-04 | 92.99 ± 8.9e-04 | 91.70 ± 6.4e-04 | 86.93 ± 1.1e-04 |
| MTO - Gradient-based | | | | | | | | | |
| CAGrad [37] | 91.25 ± 2.2e-04 | 86.79 ± 1.7e-03 | 92.88 ± 7.1e-05 | 85.12 ± 1.4e-04 | 95.46 ± 4.6e-04 | 98.17 ± 1.1e-04 | 92.91 ± 2.1e-04 | 91.74 ± 3.6e-05 | 86.88 ± 1.5e-04 |
| GradDrop [8] | 91.29 ± 2.3e-04 | 87.01 ± 5.0e-04 | 92.80 ± 1.0e-04 | 85.22 ± 8.0e-04 | 95.50 ± 2.7e-04 | 98.15 ± 6.0e-04 | 93.00 ± 9.1e-04 | 91.65 ± 1.5e-04 | 86.97 ± 4.0e-04 |
| PCGrad [64] | 91.21 ± 3.9e-04 | 86.55 ± 2.3e-03 | 92.81 ± 9.3e-04 | 85.15 ± 8.4e-04 | 95.44 ± 9.3e-04 | 98.16 ± 5.0e-04 | 92.92 ± 2.8e-04 | 91.75 ± 1.3e-03 | 86.87 ± 4.8e-04 |

Table D: (Table best seen zoomed in PDF) Results of MTL when training on all 8 tasks (subset of attributes) of CelebA for a depth of 9 layers. For PBT and PB2 we use slightly different parameters than DomainNet to account for the fact that CelebA contains more tasks, and hence has a larger search space: All PBT runs use a population size of $N = 12$ models, such that every $E_{ready} = 3$ epochs, $Q = 40\%$ of the population triggers an exploit/explore step. For PB2 runs we use a population size of $N = 8$ and otherwise the same $Q$ and $E_{ready}$ hyperparameters.

**ViT-S/4, 9 layers, 0.25 width**

| | average | age | clothes | face structure | facial hair | gender | hair color | hair style | mouth |
|---|---|---|---|---|---|---|---|---|---|
| **Scalarization** | | | | | | | | | |
| Uniform | 91.00 ± 2.5e-04 | 87.24 ± 1.6e-03 | 92.40 ± 7.6e-05 | 84.85 ± 4.5e-04 | 95.34 ± 5.7e-04 | 97.77 ± 5.0e-04 | 92.68 ± 3.5e-04 | 91.31 ± 5.8e-04 | 86.40 ± 8.0e-05 |
| PBT | 90.97 ± 1.7e-04 | 86.96 ± 1.1e-03 | 92.34 ± 2.5e-04 | 84.91 ± 8.4e-04 | 95.30 ± 8.1e-05 | 97.76 ± 2.9e-05 | 92.55 ± 1.1e-04 | 91.35 ± 6.3e-05 | 86.59 ± 2.1e-04 |
| PB2 | 91.04 ± 2.7e-04 | 87.22 ± 1.9e-03 | 92.40 ± 2.5e-04 | 84.96 ± 4.2e-04 | 95.35 ± 3.5e-04 | 97.70 ± 4.0e-04 | 92.67 ± 1.4e-04 | 91.36 ± 3.1e-04 | 86.67 ± 4.8e-04 |
| **MTO - Loss-based** | | | | | | | | | |
| Uncertainty [26] | 91.01 ± 2.3e-04 | 87.18 ± 1.4e-03 | 92.40 ± 4.5e-04 | 84.87 ± 3.5e-04 | 95.34 ± 6.6e-04 | 97.81 ± 5.7e-04 | 92.70 ± 3.9e-04 | 91.30 ± 4.5e-04 | 86.44 ± 1.1e-04 |
| IMTL-L [38] | 91.00 ± 2.4e-04 | 87.18 ± 1.5e-03 | 92.40 ± 4.6e-04 | 84.87 ± 4.0e-04 | 95.34 ± 6.6e-04 | 97.81 ± 5.0e-04 | 92.70 ± 3.3e-04 | 91.30 ± 4.7e-04 | 86.44 ± 6.2e-05 |
| **MTO - Gradient-based** | | | | | | | | | |
| CAGrad [37] | 91.07 ± 1.3e-04 | 87.19 ± 4.3e-04 | 92.55 ± 3.5e-04 | 85.03 ± 4.8e-04 | 95.41 ± 2.7e-05 | 97.79 ± 6.0e-04 | 92.82 ± 1.8e-04 | 91.52 ± 3.5e-04 | 86.23 ± 2.3e-04 |
| GradDrop [8] | 90.83 ± 1.8e-04 | 86.91 ± 4.6e-04 | 92.23 ± 1.4e-04 | 84.66 ± 1.0e-03 | 95.26 ± 6.1e-04 | 97.66 ± 3.2e-04 | 92.58 ± 3.9e-04 | 91.14 ± 2.8e-04 | 86.18 ± 4.6e-04 |
| PCGrad [64] | 90.95 ± 2.4e-04 | 87.13 ± 1.2e-03 | 92.29 ± 4.2e-04 | 84.77 ± 4.7e-04 | 95.32 ± 3.6e-04 | 97.72 ± 3.5e-04 | 92.70 ± 8.0e-05 | 91.21 ± 1.6e-04 | 86.44 ± 1.3e-03 |

**ViT-S/4, 9 layers, 0.5 width**

| | average | age | clothes | face structure | facial hair | gender | hair color | hair style | mouth |
|---|---|---|---|---|---|---|---|---|---|
| **Scalarization** | | | | | | | | | |
| Uniform | 91.32 ± 3.2e-04 | 87.26 ± 2.3e-03 | 92.81 ± 2.4e-04 | 85.19 ± 9.4e-05 | 95.51 ± 3.4e-04 | 98.15 ± 7.1e-04 | 93.03 ± 8.9e-06 | 91.66 ± 4.0e-04 | 86.93 ± 7.3e-04 |
| PBT | 91.36 ± 2.3e-04 | 87.45 ± 1.5e-03 | 92.86 ± 6.5e-05 | 85.31 ± 5.2e-04 | 95.50 ± 2.8e-04 | 97.98 ± 2.2e-04 | 93.03 ± 2.4e-04 | 91.70 ± 5.4e-04 | 87.06 ± 5.4e-04 |
| PB2 | 91.36 ± 1.4e-04 | 87.45 ± 7.8e-04 | 92.82 ± 2.7e-04 | 85.17 ± 3.1e-04 | 95.55 ± 4.4e-04 | 98.14 ± 3.4e-04 | 93.06 ± 1.2e-04 | 91.76 ± 4.3e-04 | 86.89 ± 1.7e-04 |
| **MTO - Loss-based** | | | | | | | | | |
| Uncertainty [26] | 91.33 ± 1.4e-04 | 87.32 ± 4.3e-04 | 92.84 ± 2.0e-05 | 85.19 ± 2.1e-04 | 95.52 ± 4.9e-04 | 98.15 ± 7.8e-04 | 92.96 ± 2.0e-04 | 91.69 ± 6.5e-05 | 86.93 ± 2.5e-04 |
| IMTL-L [38] | 91.29 ± 1.5e-04 | 87.25 ± 5.0e-04 | 92.79 ± 2.0e-04 | 85.17 ± 7.1e-05 | 95.50 ± 4.9e-04 | 98.10 ± 8.5e-04 | 92.97 ± 1.4e-04 | 91.66 ± 4.8e-04 | 86.92 ± 9.7e-05 |
| **MTO - Gradient-based** | | | | | | | | | |
| CAGrad [37] | 91.37 ± 1.8e-04 | 87.42 ± 2.1e-04 | 92.90 ± 4.1e-05 | 85.24 ± 1.3e-03 | 95.53 ± 2.3e-04 | 98.20 ± 1.1e-04 | 92.98 ± 2.6e-04 | 91.78 ± 2.7e-04 | 86.88 ± 3.4e-04 |
| GradDrop [8] | 91.27 ± 1.8e-04 | 87.50 ± 7.1e-05 | 92.71 ± 3.5e-05 | 85.10 ± 1.1e-04 | 95.46 ± 6.2e-05 | 98.04 ± 1.3e-03 | 92.94 ± 4.3e-04 | 91.57 ± 3.2e-04 | 86.85 ± 1.9e-04 |
| PCGrad [64] | 91.29 ± 1.6e-04 | 87.10 ± 3.2e-04 | 92.83 ± 4.3e-04 | 85.18 ± 6.1e-04 | 95.47 ± 1.1e-04 | 98.11 ± 6.4e-04 | 93.01 ± 8.9e-05 | 91.67 ± 6.3e-04 | 86.96 ± 2.6e-04 |

**ViT-S/4, 9 layers, full width**

| | average | age | clothes | face structure | facial hair | gender | hair color | hair style | mouth |
|---|---|---|---|---|---|---|---|---|---|
| **Scalarization** | | | | | | | | | |
| Uniform | 91.17 ± 1.7e-04 | 87.33 ± 5.7e-04 | 92.50 ± 8.1e-04 | 85.10 ± 4.2e-04 | 95.45 ± 1.9e-04 | 97.93 ± 1.1e-04 | 92.85 ± 2.9e-04 | 91.39 ± 1.2e-04 | 86.80 ± 6.9e-04 |
| PBT | 91.15 ± 2.7e-04 | 87.43 ± 3.1e-04 | 92.51 ± 3.6e-04 | 85.18 ± 1.1e-04 | 95.46 ± 3.1e-04 | 97.78 ± 1.5e-03 | 92.51 ± 1.4e-04 | 91.36 ± 8.0e-04 | 87.00 ± 5.8e-04 |
| PB2 | 91.25 ± 2.3e-04 | 86.83 ± 1.2e-03 | 92.85 ± 4.8e-04 | 85.12 ± 7.3e-04 | 95.49 ± 5.6e-04 | 98.19 ± 6.0e-04 | 92.90 ± 5.6e-04 | 91.78 ± 4.5e-04 | 86.81 ± 3.7e-04 |
| **MTO - Loss-based** | | | | | | | | | |
| Uncertainty [26] | 91.17 ± 1.8e-04 | 87.36 ± 6.4e-04 | 92.50 ± 8.0e-04 | 85.11 ± 3.0e-04 | 95.45 ± 1.6e-04 | 97.93 ± 1.4e-04 | 92.84 ± 3.8e-04 | 91.39 ± 5.3e-05 | 86.81 ± 8.3e-04 |
| IMTL-L [38] | 91.18 ± 1.6e-04 | 87.37 ± 4.3e-04 | 92.50 ± 8.0e-04 | 85.11 ± 2.8e-04 | 95.45 ± 1.7e-04 | 97.94 ± 2.1e-04 | 92.84 ± 3.7e-04 | 91.39 ± 6.5e-05 | 86.81 ± 7.8e-04 |
| **MTO - Gradient-based** | | | | | | | | | |
| CAGrad [37] | 91.21 ± 1.4e-04 | 87.34 ± 4.3e-04 | 92.64 ± 3.7e-04 | 85.16 ± 4.0e-04 | 95.43 ± 4.3e-04 | 97.91 ± 1.1e-04 | 92.89 ± 4.3e-04 | 91.58 ± 3.6e-04 | 86.70 ± 5.5e-04 |
| GradDrop [8] | 91.20 ± 1.4e-04 | 86.55 ± 2.1e-04 | 92.81 ± 2.6e-04 | 85.19 ± 4.3e-04 | 95.40 ± 8.1e-04 | 98.09 ± 4.6e-04 | 92.91 ± 2.6e-04 | 91.71 ± 3.4e-04 | 86.90 ± 8.9e-06 |
| PCGrad [64] | 91.13 ± 3.5e-04 | 87.35 ± 2.4e-03 | 92.50 ± 1.1e-03 | 85.04 ± 5.3e-04 | 95.40 ± 3.5e-04 | 97.87 ± 0.0e+00 | 92.82 ± 2.8e-04 | 91.37 ± 1.2e-04 | 86.71 ± 5.0e-04 |

