# OpenReview forum: "Scalarization for Multi-Task and Multi-Domain Learning at Scale"
_NeurIPS.cc/2023/Conference — NeurIPS 2023 poster_

### Official Review · Reviewer_kG9J · 2023-06-25

**Soundness:** 3 good
**Presentation:** 3 good
**Contribution:** 3 good
**Rating:** 6
**Confidence:** 4

**Summary:**

The paper studies scalarization for multi-task and multi-domain learning, which is a method to combine the losses of different tasks/domains. The authors conduct substantial experiments to draw insights into the effect of scalarization weights on multi-task/domain learning. They also propose an efficient method to search for a good set of weights.


**Strengths:**

The authors make a valuable attempt to understand the scalarization of multi-task/domain learning. It is an important problem in the literature as despite the existence of various automatic weight selection methods, it is unclear under what circumstances these methods will outperform a static scalarization. The insights are critical and novel, especially the one about conflict gradient and scalarization v.s. dynamic weight update, which is different from common belief.


**Weaknesses:**

The writing can be improved. The goal of this work is ambitious because the authors attempt to study the effect of scalarization from a wide range of aspects. However, such ambition also makes the paragraphs very condensed and jumping. In many cases, the authors bring up several insights in one paragraph with little logical flow (e.g. sec 4.1). It is better to use latex paragraph if many parallel information needs to be conveyed.

Some conceptual analysis is lacking. Since this paper is purely empirical, the generalizability of the insights is questionable if proper conceptual analysis is missing. For instance, in fig. 3, based on the two plots, it is difficult to reach any conclusions about the influence of conflicting gradient on performance because the two experiments themselves have conflicting results. For this part, an important question remained to be answered is that how conflicting gradient affect training and why in some cases it has less effects. There are a few “jump to conclusions” situations especially in sec. 4.1 and 4.2.


**Questions:**

1. In section 4.1, to obtain the optimal scalarization weight, do you perform grid search? I wonder how you are able to select a single “optimal” weight since firstly, it is impossible to search for the weight exhaustively, and secondly, it is likely that several sets of weights have undistinguishable results. Could you clarify this?
2. In fig. 4(b), it seems that training on a single task is always better than training the two tasks jointly regardless of model selection and weight tuning. Any insights on that?
3. For sec. 5, if possible, it would be interesting to see how much gap PBT closes compared to some oracle (more exhaustive grid search), even on datasets with few tasks/domains. It is unclear how good a performance a static weighting strategy can achieve because the only static baseline is uniform.


**Limitations:**

The authors have adequately addressed the limitation.

---

> ### Author Rebuttal · Authors · 2023-08-09
>
> **Improved writing:** We will carefully revise the manuscript to ensure that the logical progression of ideas is clear and coherent, and emphasize the most important insights.
>
> **Generalizing the conclusions of Figure 3:** We expanded the results of Figure 3, as detailed in the attached PDF to the global response (Figure 1). This extended analysis includes more tasks and varying learning rates to convey our insight more clearly: the MDL/MTL models with the highest generalization performance are not necessarily the ones with the least amount of gradient conflicts, and vice versa. We also observe that, while the global amount of gradient conflicts tends to increase with a higher number of tasks, the overall trend of each curve is rather consistent across model sizes.
>
> **Question 1 (optimal scalarization weight in analysis)** Indeed, for Figure 4.1, our approach essentially involves a grid search: For each setting, we perform a sweep for the task weight over $p_{task_1} \in \{0.1, 0.2, 0.3, 0.4, 0.5, 0.6, 0.7, 0.8, 0.9\}$ and then report the relative improvement of the MTL/MDL model over the corresponding single task baseline for the best ratio (where the best ratio is defined as the one that yields maximum accuracy averaged across the two tasks/domains): This can essentially be interpreted as having an oracle to select the task weight. In Section 2.1 of the supplemental material we also report some plots illustrating the results for all the ratios $p_{task_1}$ we sweep over.
>
> **Question 2 (Figure 4b):** Indeed, on the Taskonomy dataset we sometimes observe that the STL baseline outperforms the MTL models still. While this could simply be a symptom of the specific architecture/hyperparameters we considered, we also hypothesize that this might show the limits of forcing all tasks to share a common encoder: In fact, the Taskonomy dataset was initially introduced as a benchmark to uncover groups of tasks that can benefit from training together, from ones that should be kept fully separate to avoid interference [f, g]. Building on this orthogonal line of work, we posit that some task interferences may only be resolved through explicit architectural modifications. For example, allocating dedicated encoders for specific tasks might mitigate the observed performance discrepancy.
>
>
> **Question 3 (PBT vs static grid search oracle)**
> We address this question from both theoretical and practical perspectives:
>   * **In theory**: While PBT itself does not provide any guarantee, its more recent variant PB2 [e] (which uses Bayesian Optimization to guide the exploration) provides a theoretical regret bound as a function of the population size $N$.
>   * **In practice**: In Table 2 of the attached global response PDF, we performed an experiment for comparing grid search (with uniformly distributed grid points) to PBT in a small setting with only three tasks/attributes of CelebA (`Five_o_Clock_Shadow`, `Arched_Eyeberows` and `Attractive`). Due to time constraints, we were only able to run experiments up to using 5 (uniformly distributed) grid points per task weight for the grid search, resulting in 125 models to run. In Table 2a, we observe that grid search can outperform the dynamic search of PBT, but with a much higher computational budget.
> We further illustrate the search space covered by PBT during its dynamic search in Figure 3b: each point corresponds to a 3-scalarization weights configuration encountered during the dynamic PBT search. This visualization contrasts PBT's exploration pattern with the regular uniform grid used in a classic grid search, providing insights into how the two methods differ in navigating the search space.
>
> **References**
>   * [e] Provably Efficient Online Hyperparameter Optimization with Population-Based Bandits, Parker-Holder et al
>   * [f] Which Tasks Should Be Learned Together in Multi-task Learning? , Standley et al
>   * [g] Disentangling Task Transfer Learning, Zamir et al

---

> > ### Comment · Reviewer_kG9J · 2023-08-17
> >
> > I want to thank the authors for the detailed feedback, which addresses most of my concern. The newly included results on PBT v.s. grid search in the global response PDF provides extra insight for the problem. Hence, I will raise my score to 6.

---

### Official Review · Reviewer_ytBJ · 2023-07-07

**Soundness:** 2 fair
**Presentation:** 3 good
**Contribution:** 3 good
**Rating:** 7
**Confidence:** 4

**Summary:**

This paper seeks to better understand the complexities of unitary scalarization for multitask and multi-domain learning. The paper explores the impact of model size, degree of gradient conflict and variations in scalarization weights in order to derive a set of guiding principles for MTL/MDL.
The authors also propose to leverage existing population based HP optimization procedures to efficiently search for the best set of scalarisation weights

**Strengths:**

1. The paper presents an expansive set of experiments to understand the impact of model size, multi-task setting (MTL vrs MDL) and task affinity on the performance of the unitary scalarization approach
2. The paper is actionable -- it proposes to leverage pre-existing population based HP optimization approaches to search for the best scalarization weights.
3. The paper is well written and the experimental methodology is described in sufficient (reproducible) detail


**Weaknesses:**

My main issue with the paper is that I am hesitant about specific parts of the experimental methodology.

Primarily, in  section 4, the experimental procedure is described as tuning HP for single task and then using the best single task HP for all followup multitask experiments. This creates an unfair comparison since it assumes that the HP setting that is best for the single task is also optimal for the MT setting thus bringing the robustness of the results to question. (This is especially considering that, forcing the scalarization weights to sum to 1, means that the effective per-task learning rate is always smaller than for the single task setting)

Also, there exists confounding variables for the the experiments on **model capacity and  gradient conflict**  that are not addressed.
1. Are the models used ResNet models used for Figure 3  pre-trained or learned from scratch ?
2.  How were the learning rates for this experiment chosen ?  In general, I suspect that there is also a dependence of the degree of conflict (after a reasonable number of epochs like 1 in your case) on the learning rate used. It would be important to see if the degree of conflict (at a fixed model size) -- varies substantially with learning rate and whether this variance is smaller or larger than the variance that comes changing model size at fixed LR.


----- Update ------

Updated score after rebuttal. Thanks for the responses @ Authors

I am willing to raise my score if these concerns are addressed.

Missing relevant citation
1. Exploration of model capacity and scalarization weight https://arxiv.org/pdf/2302.09650.pdf


**Questions:**

Questions
1. Is the sweep to find the best single dataset HP performed for each model size (For figure 2) or is it performed for 1 model size and then used across all sizes ?
2. For figure 3, how do the hyper-parameters like learning rate differ from for model size
    1. Do you use a fixed learning rate across all model sizes ? Or use a pre-determined best learning rate for each model size ?
3. Are the resnets in Figure 3 pre-trained resnets ?
    1. It would be interesting to see if the effect still holds for resnets trained from scratch vrs pre-trained on say Imagenet


**Limitations:**

Authors have addressed limitations.

---

> ### Author Rebuttal · Authors · 2023-08-09
>
> **Gradient conflict and impact of learning rate:** Following reviewer ytBJ's suggestion, we added gradient conflict measurements experiments with varying learning rates, as illustrated in Figure 1 of the attached PDF (global response). The results show that the variance of gradient conflict measurements tends to be higher across learning rates than across the different model sizes we considered. Notably, we observe that when the learning rate is excessively high, leading to divergence in training loss, there are distinct peaks in gradient conflicts. Nevertheless, when the loss is well-behaved (does not diverge), the general trend of gradient conflict measurements across training epochs remains similar across learning rates.
>
> **Learning rate choice:**
>   *  **For the different model sizes in the analysis:** We do conduct a sweep across different learning rates for each single task architecture (e.g. `[3e-1, 3e-2, 3e-3, 3e-4]` for DomainNet). However, in practice, we found that the optimal single-task learning rate remained consistent across different model sizes. While this might vary with a more fine-grained learning rate grid search or different family of architectures, our analysis led us to a specific learning rate for one model with different depths/widths for practical purposes. In practice, the only parameter we adapt for different model sizes is the batch size and the number of gradient accumulation steps.
>   * **PBT experiments:** For the experiments of section 5 in the paper (PBT and MTO comparison), we train all models for different learning rates and report results for the best performing one (specifically the sweep is done over `[5e-5, 5e-4, 5e-3]` for CelebA and `[3e -3, 3e-2]` for DomainNet)
> * **Normalizing scalarization weight to sum to 1:**  We acknowledge that removing the constraint of normalizing scalarization weight to sum to 1 could be akin to further tuning the learning rate for each MTL model, potentially leading to improvements over the single task baseline. Since our primary goal was to compare the performance of the different MTL/MDL against each other (mainly using single-task performance to compute relative improvement), we chose to keep the normalization constraint to keep the search space at a reasonable size and make the analysis more scalable.
>
> **Pretrained vs from scratch models:** In all experiments, we train the models from scratch. It is very likely that different pretraining strategies would impact task interference and MTL/MDL performance, but we didn't investigate this angle in this work.
>
> **Missing citation:** Thank you for the suggestion. We will add the citation on designing scaling laws for multi-lingual language models in the related work section

---

> > ### Comment · Reviewer_ytBJ · 2023-08-16
> > **Question about PDF uploaded**
> >
> > Hi Authors,
> > Thank you very much for your responses.
> > I have quick question about Fig 1(a) in the PDF that you uploaded. Since the model sizes are not marked on the line, it is really hard to see what is going on here.
> >
> > I guess the question I was trying to have answered with the learning rate vrs capacity question is this : at any fixed point in training (say 50%),  if we consider two learning rates $l_1$,  $l_2$ that are sufficiently different but not divergent (w.r.t the model), and we consider   $\mathrm{model}_1$, $\mathrm{model}_2$ where size(model1) < size(model2), could
> >
> >  $$\text{gradconflict}(\mathrm{model}_1, l_1) <  \text{gradconflict}(\mathrm{model}_2, l_2)$$
> >
> > but
> >
> >  $$\text{gradconflict}(\mathrm{model}_2, l_2) <  \text{gradconflict}(\mathrm{model}_1, l_2)$$
> >
> >
> > This would mean that the conclusion that larger models have higher conflict would be invalid except when conditioned on a specific choice of learning rate.

---

> > > ### Author Response · Authors · 2023-08-16
> > > **Impact of the learning rate on the relative model sizes' ranking wrt. gradient conflicts**
> > >
> > > Hello reviewer ytBJ,
> > > thanks for your response and for clarifying the question. Please find our answers below
> > >
> > > **1.** We did not claim that larger models always imply more gradient conflicts, sorry if the text was misleading in that regard. Rather our main observation regarding model capacity was that changing model capacity does not significantly impact the magnitude of gradient conflicts, and yet it does have a visible impact on the MTL/MDL performance (e.g. line 216 and Figure 3a).
> > >
> > > **2.** Following your question, we looked further into whether the relative ordering of model sizes based on gradient conflicts changes with learning rate. Our methodology was as follows:
> > >   * We take the data from Figure 1a and rank each model size in terms of gradient conflict, for each learning rate and time step (in ascending order, rank of 1 = lowest gradient conflict).
> > >   * Across time steps, we compute the most common rank, as well as how many times the rank at any time step matches the most common one (we call this ratio `consistency`) for each model size and learning rate
> > >   * We report these values (most common rank and consistency across time steps) for different model sizes pairs and learning rate
> > >
> > > **summary:** Our main observation is that generally, larger models do exhibit more gradient conflicts (higher global rank), but the consistency of this behavior indeed is impacted by the learning rate: the relative ranking fluctuates more at lower learning rates (lower consistency)
> > >
> > > ### DomainNet - 6 tasks - ResNets
> > > |          |   r26 |   r50 | consistency   |
> > > |:---------|------:|------:|:--------------|
> > > | lr=0.003 |     1 |     2 | 56.7%         |
> > > | lr=0.03  |     1 |     2 | 73.3%         |
> > > | lr=0.3   |     1 |     2 | 93.3%         |
> > >
> > >
> > > ### CelebA - 40 tasks - ViT-S/4 - fixed depth
> > >
> > > |           |   w=0.5, d=3 |   w=1, d=3 | consistency   |
> > > |:----------|-------------:|-----------:|:--------------|
> > > | lr=0.0005 |            1 |          2 | 52.0%         |
> > > | lr=0.005  |            1 |          2 | 74.0%         |
> > >
> > > |           |   w=0.5, d=9 |   w=1, d=9 | consistency   |
> > > |:----------|-------------:|-----------:|:--------------|
> > > | lr=0.0005 |            1 |          2 | 66.0%         |
> > > | lr=0.005  |            1 |          2 | 70.0%         |
> > >
> > >
> > > ### CelebA - 40 tasks - ViT-S/4 - fixed width
> > > |           |   w=1, d=3 |   w=1, d=9 | consistency   |
> > > |:----------|-----------:|-----------:|:--------------|
> > > | lr=0.0005 |          1 |          2 | 60.0%         |
> > > | lr=0.005  |          1 |          2 | 58.0%         |
> > >
> > > |           |   w=0.5, d=3 |   w=0.5, d=9 | consistency   |
> > > |:----------|-------------:|-------------:|:--------------|
> > > | lr=0.0005 |            2 |            1 | 52.0%         |
> > > | lr=0.005  |            1 |            2 | 70.0%         |

---

> > > > ### Comment · Reviewer_ytBJ · 2023-08-17
> > > > **Thank you - updated paper score**
> > > >
> > > > Hi Authors,
> > > > thanks for the clarification -- I have updated the score of the paper to reflect the updates / clarifications.

---

### Official Review · Reviewer_wCAe · 2023-07-07

**Soundness:** 3 good
**Presentation:** 4 excellent
**Contribution:** 2 fair
**Rating:** 6
**Confidence:** 4

**Summary:**

This work is interested in analyzing the extent to which scalarization is an effective strategy against negative transfer in multi-task and multi-domain learning. Scalarization focuses on selecting an adequate weights for a convex combination of task losses, rather than employing expensive or complex conflict mitigation strategies. Although the search space for scalarization weights grows exponentially with the number of tasks, during training minimizing a weighted sum of task losses is fast and simple compared to many complex optimization methods and has recently been shown to be just as good.

This work therefore attempts to better understand the dynamics of scalarization in multi-task models by studying multi-task generalization under scalarization alone. They find a few findings which appear consistent across the settings they consider: the first is that scalarization is more effective as model capacity grows; the second is that uniform scalarization is rarely optimal, implying that for each MTL setting the scalarization weights must be tuned; the final observation is that while gradient conflict can predict e.g. task affinity, it does not predict generalization because model capacity does not observably affect gradient conflict.

Using these observations, the authors posit that a strong scalarization approach to MTL can be extremely effective, but the search for the optimal scalars makes it more expensive than other proposed methods. To this end, the authors leverage population based training to efficiently explore the parameter space of task weights. They find that models trained with scalarization weights from PBT outperform the uniform scalarization baseline, as well as several other sota optimization methods.

**Strengths:**

- The paper is well written. It conveys its core ideas and motivation clearly.
- The in-depth, rigorous exploration into multi-task learning dynamics is important as recent work has shown that most prior optimization work is not actually beneficial for standard MTL problems.
- The observation that uniform weighting is rarely optimal is useful, even if not surprising.
- The findings w.r.t. task conflict and generalization are very interesting, and helpful to consider in further development of MTL methods.
- In total, the analysis of section 4 could be helpful for the design of future methods which aim to target scalarization, and they serve to motivate the proposed method in section 5.
- The proposed method is clearly useful empirically, and demonstrates the effectiveness of scalarization vs. other, much more complex, optimization methods.

**Weaknesses:**

- All 3 key conclusions come from experiments which study only 2 tasks at once. While it is not unreasonable to extrapolate some conclusions from this setting, some conclusions could be at least verified for larger task settings, even up to 3 or 4 simultaneous tasks just to ensure the trends still hold. For example, does model capacity really not affect gradient conflict levels when considering all 40 tasks of CelebA?
- The models all use the optimal single-task parameters but this might be unfair to the MTL models, e.g. [1] suggested that the learning rate should scale with the number of tasks if all else is fixed.
- The final results should probably use random scalarization [2] as an additional baseline. I find this to be especially true given that the uniform models almost uniformly outperform the other optimization methods, so it is not extremely surprising that additional tuning of the task weights will result in the best performance on the tables.
- The comparison to previously considered SOTA methods only goes up to 7 or 8 tasks, whereas many of the methods were tested on e.g. up to all 40 tasks on CelebA. It’s not clear if PBT can efficiently scale up to 40 tasks. To that end, a comparison of the overall compute used by the tested methods would be really helpful.

[1] the importance of temperature in multi-task optimization, Mueller et al., 2022

[2] reasonable effectiveness of random weighting: a litmus test..., Lin et al., 2021

**Questions:**

- Do you know how PBT scalarization compares to random scalarization?
- How does the entire procedure of PBT compare to other optimization methods w.r.t. total compute time or flops?
- Finally, I’m particularly interested in whether or not the trends w.r.t. model capacity and gradient conflict hold as the number of tasks increases. Do you happen to know if they do?

**Limitations:**

yes

---

> ### Author Rebuttal · Authors · 2023-08-09
>
> **1. Gradient conflicts:** In response to reviewer wCAe's inquiry about gradient conflicts, we have expanded our analysis to include experiments that encompass all 40 tasks/attributes of CelebA and all 6 domains of DomainNet, when training a uniform MTL model. These additional experiments are illustrated in Figure 1 of the attached PDF. Generally, we observe that while the proportion of gradient conflicts may increase with the number of tasks, the overarching trend remains consistent: Low gradient conflict does not necessarily correlate with optimal MTL/MDL performance. This observation holds true across different model sizes and aligns with the conclusion C2 of the submission.
>
> **2. Population Based Training (PBT)**:
>
> **a. Computational cost:** A nice feature of Population-based Training search is that the computational cost can be easily decorrelated from the number of tasks. In fact, the main computational cost comes from the number of models in the population ($N$): While increasing $N$ does increase the search space coverage, it is also possible to control the exploration/exploitation trade-off through other parameters; namely the number of epochs before models in the population pause and are compared against one another ($E$) and the proportion of the population killed in each exploration step ($Q$); both $E$ and $Q$ have an impact on the computational cost which is often negligible in practice as it only incurs a few additional checkpointing/writing operations. In other words, even for a large number of tasks, we can use a low value of $N$ which introduces a natural trade-off between the compute budget allocated to the scalarization weight search and search space coverage.
>
>   To better highlight this trade-off, we report additional results of PBT search while varying the hyperparameters $N$ and $E$ in the global response (Table 1a); As suggested by reviewer  wCAe, we scale the experiments to cover all 40 tasks on CelebA. In that setting, while increasing population size generally improves the search result, we observe that *(i)* we can get good performance even when the population size is significantly smaller than the number of tasks and *(ii)* in the regime of a large population, the cost of going from e.g. N=24 to N=40 is not worth the gain in accuracy.
>
> **b. Comparison to multi-task optimization methods (MTO):** In the attached PDF (Table 1b), we summarize the computation costs of PBT and some MTO methods. The key difference between gradient-based MTO and PBT lies in the memory bottleneck: PBT requires training $N$ models, but independently, hence the algorithm does not require additional storage compared to training a single model. In contrast, gradient-based MTO methods only train a single model, but require storing each per-task gradient to compare them against each other in every training iteration.
>
> Nevertheless, the efficiency comparison between MTO methods and PBT search is highly dependent on available resources. For scenarios with ample RAM/GRAM and a reasonable number of tasks, MTO methods may still be computationally viable. Conversely, PBT's natural parallelism is favorable in scenarios with many memory-constrained devices.
>
>
> **3. STL vs MTL learning rate**
>
> **a. PBT experiments**: For the experiments of section 5 in the paper (PBT and MTO comparison), we did train all models for different learning rates and report results for the best performing one (specifically `[5e-5, 5e-4, 5e-3]` for CelebA and `[3e -3, 3e-2]` for DomainNet)
>
> **b. Analysis**: Indeed, further tuning the learning rate for MTL models could lead to better performance with respect  to the STL baseline; in our setting, this would be roughly equivalent to dropping the constraint that the scalarization weights have to sum to 1 (line 115) but would also greatly increase the search space / number of experiments.
> 	In our analysis, we mainly use the single task as a reference baseline with the primary goal to compare the relative improvement of the different MTL/MDL against each other (for different task ratios and model sizes). This is why we chose to select the learning rate as to favor the STL baseline rather than a specific MTL setting (e.g. the uniform weights).
> 	Nevertheless, after reviewing the suggested reference [1], it would indeed be interesting to investigate how their conclusions on optimal learning rate/task temperature evolve across different model sizes.
>
> **Random scalarization baseline**. We added results for the suggested random scalarization baseline (RLW) of **[2]** for Table 2a in the attached  global response PDF for the CelebA setting with 40 tasks. Following the reference, in every training iteration we sample task weights from N(0, 1) and normalize them via softmax function. In that setting, we observe that RLW yields a slightly stronger baseline than uniform weighing on average, and that the configuration found by PBT outperforms both when searching with more than $N=6$ population size.

---

> > ### Comment · Area_Chair_pbJY · 2023-08-21
> > **Thanks for the rebuttal.**
> >
> > Although the review did not engage, I'll carefully read and consider it during the decision period.
> >
> > AC

---

> > ### Comment · Reviewer_wCAe · 2023-08-21
> >
> > Thank you for your response. You have addressed my two key concerns (limited analysis to 2 tasks and comparing to RLW), and so I will raise my score to a 6!

---

### Official Review · Reviewer_iXKX · 2023-07-11

**Soundness:** 3 good
**Presentation:** 2 fair
**Contribution:** 1 poor
**Rating:** 3
**Confidence:** 4

**Summary:**

This paper analyzes multi-task learning (MTL) and multi-domain learning (MDL) setting. The paper observes several points: 1. MDL/MTL improvements are more significant with bigger network capacity, 2. Gradient conflicts are not necessarily well correlated with MDL/MTL performances, 3. Tuning scalarization weight is important for MDL/MTL performance, 4. population-based training (PBT) can be an efficient way to tune the scalarization weights when there are many tasks. The experimental results support each of their claims, and lastly this paper show that PBT can be even competitive with memory-expensive gradient-based methods, such as PCGrad.

**Strengths:**

- Paper was easy to read.

**Weaknesses:**

[major comments]

- overall, almost all observations found by this paper seem quite trivial to me. Currently, I don't think this paper provides very useful insights or something that has not been investigated before.
- For instance, in page 2, (C1) is already quite trivial - we already know that larger network capacity can mitigate negative interference because larger capacity means it can accommodate more diverse information. See [1] for the reference.
- (C3) is also trivial - we already know that tuning scalarization weights is important and they should be tuned differently for each task/domain/architecture, and so on.
- The conclusion of (C2) is misleading, in my opinion. The authors observed that MTL/MDL performance improves with bigger network capacity while the degree of gradient conflict remains the same, and they conclude that gradient conflict do not correlate well with the actual MTL performance in practice (L219-220). This conclusion sounds weird because the network architecture is different. What if the network architecture remains the same and we resolve the gradient conflict, which is the usual assumption of other papers, such as PCGrad?
- (C4) is simply an application of the existing technique (PBT) to scalarization weights, which is not very surprising. And the authors did not provide other baselines than Uniform, although there should be many existing methods that allows to carefully tune the scalarization weights.
- The same for the conclusion in section 4.1. All (C1), (C2), and (C3) sound obvious to me.
- (C1): we already know that larger network capacity can mitigate negative interference effect, as mentioned above.
- (C2): Of course the best-performing scalarization weight would not be p1=p2=0.5.
- (C3): Of course MTL/MDL has a regularization effect, so the training loss converges slower but the test accuracy is higher.

[minor comments]

- in (1), the notation $\frac{\nabla}{\nabla}$ looks very weird. It should be either $\frac{\partial}{\partial}$ or simply $\nabla_\theta \mathcal{L}$.
- in (2), the definition of $f$ is missing. What is it? (I assume that it's $\nabla_{\theta_i} \mathcal{L}_t(x_t,y_t)$?)
- missing baseline - Sequential Reptile [2], which can resolve gradient conflict issue without heavy memory overhead.

[reference]

[1] Wang et al., On Negative Interference in Multilingual Models: Findings and A Meta-Learning Treatment, 2020
[2] Lee et al., Sequential Reptile: Inter-Task Gradient Alignment for Multilingual Learning, 2022

**Questions:**

See the comments above. Overall, I don't think this submission is above the acceptance bar. Most of the observations seem obvious and not very informative. In order for such an analysis-style paper to be accepted, the analysis should be 1. better organized with clear insight, 2. providing novel and useful insights that have not been found by other researchers.

------------------------------------------------------------------
[After rebuttal]
I read the author's rebuttal and other reviewer's comments. Unfortunately, I'm still not convinced of the rebuttal, thus I maintain my current score.

**Limitations:**

The authors have properly addressed the limitations of this paper in Sec 6.

---

> ### Author Rebuttal · Authors · 2023-08-09
>
> **Intuitiveness is not a weakness:** While some of the observations in our paper may align with intuitive understanding, intuition does not always translate into empirical evidence. Our work aims to provide a rigorous and systematic analysis of multi-task learning (MTL) and multi-domain learning (MDL), and we believe that our findings contribute novel insights to the field. In particular, insights such as C1 and C3 are rarely taken into account or emphasized in previous multi-task optimization works, hence labeling them as "obvious" and "trivial" does not reflect current literature. Nevertheless, if reviewer iXKX would like to suggest additional references we should examine, we would be happy to further address those during the discussion period.
>
> **(C1) [Model size]**
> We are not aware of a previous reference analyzing and generalizing the link between model size and MTL/MDL performance. In fact, many MTO works (e.g. [a, b] for recent references) are still evaluated on very specific architecture/dataset combinations, omitting the effect of model size that may affect model comparison.
>
> We have reviewed the suggested reference [1], and it focuses primarily on the impact of dataset size on negative interference in multi-lingual models. While this is an important aspect, it does not directly address the link between model size and MTL/MDL performance that our paper explores.
>
> **(C2) [Gradient conflict]**
> Our conclusion (C2) does not undermine the effectiveness of existing gradient conflict resolution methods in multi-task learning (MTL) or multi-domain learning (MDL). Indeed, as highlighted in reference [c], multi-task optimization methods such as IMTL or PCGrad can have a beneficial regularization effect.
>
> However, our analysis shows that minimizing gradient conflict does not necessarily lead to optimal MTL/MDL performance. We believe this perspective adds valuable insights as it challenges the common assumption that reducing gradient conflicts to zero should be the de-facto way to solve task interference in MTL/MDL (additional figures and results can be found in the PDF attached to the global response).
>
> **(C3) [Tuning scalarization weights]**
> We do not think it is obvious that uniform weights p1=p2=0.5 are never the best-performing solution: Uniform weighing can be a reasonable assumption as MTL models are usually evaluated by taking the uniform average of their respective task metrics, in particular when the training losses and test metrics coincide (e.g., in the Taskonomy example we consider in the main paper). In fact, the use of uniform weighing for task losses remains a prevalent scalarization method in both vanilla MTL and gradient-based multi-task optimization methods.
>
> Moreover, the claim of (C3) is not solely on the benefits of tuning scalarization weights, a concept that has indeed been explored, for example, in [d]. Our work also investigates how and whether the optimal scalarization weights evolve across different model capacities (lines 56-60 and section 4.3). This exploration adds a new dimension to the understanding of scalarization in MTL/MDL. and provides insights that extend beyond existing literature.
>
> **(C4) [Application of PBT]**
> To the best of our knowledge, there does not exist a well-established efficient method to tune scalarization weights in large-scale settings: Classical search algorithms such as grid search or Bayesian optimization do not scale well to larger number of tasks/tunable parameters. While PBT itself is not a novel search algorithm, showing that it can be successfully applied to the context of scalarization is a novel contribution of our work; We additionally show that the learned schedule of dynamic task weights through PBT can compete with automatic task weighing from state-of-the-art MTO methods.
>
> In conclusion, we believe that (C4) offers valuable insights and an efficient practical solution for tuning scalarization weights.
>
>
> **(Minor comments)**
>   *  $f$ refers to an arbitrary function of $(x, y)$ to illustrate the link between the reweighing and resampling formalisms; but indeed in the context of equation (1), it can be replaced with $\nabla_{\theta_i} \mathcal{L}_t(x_t, y_t)$
>   * The suggested reference **[2]** tackles the problem of transfer learning/catastrophic learning in multilingual learning which differs from our setting (multi-task learning from scratch). It also does not incur memory overhead compares to other MTO methods, but it trades it off for additional computations (each parameter update necessitates K gradient steps in Equation (7) of **[2]**).
>
>
> **References**
>   * [a] RotoGrad: Gradient Homogenization in Multitask Learning, Javaloy et al
>   * [b] Just Pick a Sign: Optimizing Deep Multitask Models with Gradient Sign Dropout, Chen et al
>   * [c] In Defense of the Unitary Scalarization for Deep Multi-Task Learning, Kurin et al
>   * [d] Do Current Multi-Task Optimization Methods in Deep Learning Even Help?, Xin et al

---

> > ### Comment · Area_Chair_pbJY · 2023-08-21
> > **Thanks for the rebuttal**
> >
> > Although the review did not engage, I'll carefully read and consider it during the decision period.
> >
> > AC

---

### Author Rebuttal · Authors · 2023-08-09

We thank all the reviewers for their insightful and detailed feedback. We have addressed each reviewer's concerns separately through comments, and we would like to use this global response to highlight the additional experiments we mention in these responses, which are illustrated in the attached PDF. Finally, we would be happy to address any further questions during the discussion period.

**Gradient conflict (Figure 1):** To reinforce the conclusion *(C2)* (Figure 3 of the main submission), we follow the reviewers' suggestion to investigate how gradient conflict measurements evolve with respect to the number of tasks (reviewer wCAe) and the learning rate (reviewer ytBJ).  In these experiments, we follow the same protocol as in Section 4.2, by measuring the percentage of gradient conflict pairs in each training epoch. We treat each attribute in CelebA as a separate task (40 tasks total).  We vary the learning rate (in `[5e-4, 5e-3, 5e-2]`) and model size (depth in `[3, 9]`; width in `[0.5, 1]`). We believe these additional experiments reinforce our observations from Section 4 demonstrating that while high gradient conflict is indeed indicative of bad MTL/MDL performance, low gradient conflict does not correlate well with best MTL/MDL performance.

**Population-based Training:**
  * **a) Computational cost (Table 1):** To address reviewer wCAe's question, we scale the PBT search to all the 40 tasks of CelebA and report the results in Table 1a. We report results across different population sizes (N), illustrating that the population size (hence the computational cost) does not have to scale linearly with the number of tasks. In addition, in Table 1b, we present a brief overview of the theoretical computational/memory cost of PBT compared to uniform MTL and state-of-the-art MTO methods.

  * **b) Comparison to grid search (Table 2):** To address reviewer kG9J's question, we report results for classic grid search on a small-scale MTL setting consisting of three attributes/tasks of CelebA. We also illustrate how the search space covered by PBT differs from the classic uniformly distributed grid search space.

---

### Decision · Program_Chairs · 2023-09-21

**Decision:**

Accept (poster)

**Comment:**

The paper studies the intricacies of scalarization in the context of multi-task learning (MTL) and multi-domain learning (MDL). Through empirical analysis, several interesting albeit intuitive observations are made.

The paper tackles an essential problem in the ML domain and presents a rigorous empirical analysis to derive several crucial insights. Although some reviewers have raised concerns, the findings worth to be shared with the community. The authors are strongly encouraged to address the mentioned concerns before the camera ready version.